METHODS

# Automating population construction and parallel simulation of biophysical models for neuromuscular cells: An inverse approach

Hojeong Kim [1,2*]

**1** Division of Biomedical Technology, DGIST, Daegu, Republic of Korea, **2** Department of Interdisciplinary Studies, DGIST, Daegu, Republic of Korea

* hojeong.kim03@gmail.com

## Abstract

Biophysical modeling and simulation help to promote a comprehensive understanding of the neuromuscular mechanisms underlying muscle force generation and control in normal and pathological states. However, this process is labor intensive and limited to special conditions due to the heterogeneity of neuromuscular cells and the variability in their organization across body parts and ages. We present a methodology to resolve this issue. First, we formulate a building-block approach with an inverse modeling framework for automated population construction and tractable hierarchical analysis under various physiological conditions. Second, we devise a network folder-based approach with a virtual environment technique for efficient parallel simulation that can operate on a multicore computer, a supercomputing system, or a computer network through the internet. Third, we implement the methodology by developing open-source command-line software called pNMS. Finally, we demonstrate that pNMS can replicate experimental and simulation results from different environments and predict the population behaviors of neuromuscular cells depending on their organization and muscle length. With an intuitive, flexible application programming interface, this software tool may offer a solution for promoting efficient investigation and an in-depth understanding of neuromuscular function at cellular resolution under realistic scenarios.

## Author summary

Determining model parameters is challenging in biophysical modeling and simulation of neuromuscular systems at cellular resolution. Optimization of model parameters is typically performed to replicate input-output data observed in target neuromuscular cells. Yet, this forward approach becomes labor-intensive and nearly impractical as the number of neuromuscular cells increases, and their properties vary according to body parts, activity, aging, and disease. This

**Data availability statement:** All relevant data are within the manuscript and its Supporting Information files.

**Funding:** HK received funding from the National Research Foundation of Korea (NRF, https://www.nrf.re.kr) (2021R1F1A1062265) and the Ministry of Science and ICT (MSIT, https://www.msit.go.kr) (DGIST R&D Program 24-NT-01). The funders had no role in study design, data collection and analysis, decision to publish, or preparation of the manuscript.

**Competing interests:** I have read the journal's policy and the authors of this manuscript have the following competing interests: HK has patent (KOR: 10-1769779) issued to DGIST.

study proposes a methodology based on an inverse modeling framework and a network folder technique for automating cell population construction and parallel simulation under various computer systems. The methodology is implemented as an open-source command-line software tool called pNMS to promote easy use with fewer codes. We predict the population behaviors of neuromuscular cells under various their organization and muscle length. This study offers a solution for accelerating basic research on neuromuscular function, classroom education, and engineering for motor rehabilitation.

## Introduction

Motoneurons in the brainstem and spinal cord, along with their innervating muscle fibers, are the primary cells of the neuromuscular system that determine the muscle force required to generate proper movement of body part [1]. In this neuromuscular system, a few to hundreds of motoneurons directly connect to a single muscle comprising tens to thousands of muscle fibers [2]. The smallest functional structure underlying muscle force control is the motor unit, a single motoneuron and the muscle fibers it innervates [3]. Therefore, to fully understand the functioning of the neuromuscular system, both the neural and muscular mechanisms need to be investigated at the cellular level in an integrative framework.

The neural and muscular systems are typically studied separately. For motoneurons, morphological and electrophysiological experiments have revealed cell-specific properties in both passive and active mechanisms underlying firing behavior [4], as well as their systematic variations across the motoneuron population [5,6]. Similarly, for muscles, cellular and whole muscle studies have reported a range of cell-specific properties for both passive and active mechanisms mediating force generation [7–9], as well as their systematic variations across the muscle fiber population [6,10]. These studies highlighted the nonlinear behavior and heterogeneous properties of motoneurons and muscle fibers in the neuromuscular system. Furthermore, the organization of these primary cells has been shown to vary widely depending on multiple factors, such as body region [11], activity [12], aging [13], and disease [14]. However, experimentally unraveling whether and how these cellular properties contribute to behaviors at different levels of the neuromuscular system remains challenging.

Computer modeling and simulation have been employed to address this fundamental issue. Biophysically informed computational models for motoneurons can fall into two categories—full [15] and reduced [16] modeling approaches—depending on the inclusion of anatomical dendritic structures. Both modeling approaches have faithfully replicated the nonlinear firing behavior of motoneurons, incorporating cell-specific properties characterized under various experimental conditions. Moreover, reduced modeling approaches have shown greater computational efficiency and analytical analysis by applying mathematical tools such as nonlinear dynamical systems theory [16].

For muscles, mathematical modeling approaches have been used to model and simulate muscle force production at the cellular [17–19] and whole muscle [7,20,21]

levels. Cellular models based on chemical reactions have been developed for microscopic relationships between calcium dynamics, actin–myosin interactions, and force production during isometric contractions. In contrast, whole muscle models based on Hill mechanics have focused on macroscopic relationships among force, stimulation frequency, and muscle length. A modular framework unifying these muscle models has demonstrated the full spectrum of muscle fiber behaviors under physiological conditions [22,23].

Population models are typically constructed with three types of neuromuscular cells: slow (S), fast but fatigue resistive (FR), and fast but fatigable (FF) [2]. However, the intrinsic properties of motoneurons and muscle fibers have been shown to vary continuously across the neuromuscular system [3]. Furthermore, most motor unit population models have included a simple mathematical muscle model that is not appropriate for incorporating changes in the cellular properties and muscle length [24]. Hence, previous population modeling approaches should be extended to reflect the continuum variation in intrinsic cell properties observed in the neuromuscular system.

One challenge in empirically based simulations is determining model parameters that replicate various experimental observations under different experimental conditions. This issue has been addressed by numerically optimizing multiple model parameters to reproduce a set of target cell outputs in response to representative stimulation protocols [25]. However, this forward approach has raised practical issues as the number of target parameters and the amount of data increase, including significant reductions in convergence efficiency, model stability, parameter uniqueness, and simulation accuracy.

As an alternative, an inverse modeling framework has been proposed to overcome these issues by inferring and constraining effective model parameters directly from essential cell properties measured under various experimental conditions rather than cell outputs [26,27]. This inverse approach has proven useful in constructing biophysically plausible neuronal models that capture the systematic variation in cell-specific properties across the pool of spinal motoneurons [28,29]. However, population models for motoneurons remain manually constructed for a typical case and are simulated via specific software, which often requires additional programming for parallel simulation. As a result, the utility of computer modeling and simulation is limited by the current labor-intensive, time-consuming process, especially in terms of customization and analysis for various normal and abnormal situations in the neuromuscular system across body parts and ages.

To address this current issue, we propose a methodology that facilitates the development and simulation of motor behavior models of the neuromuscular system at single-cell resolution. We develop the methodology through the following steps:

1) A building-block approach combined with an inverse modeling framework is formulated to automatically construct population models and perform tractable hierarchical analyses from subcellular activities to population behaviors.

2) A network folder-based approach combined with a virtual environment technique is developed for efficient simulation in various parallel computing systems, such as a multicore computer and a high-performance computing cluster.

3) An open-source command-line software tool called pNMS is developed to verify our methodology by replicating the experimental and simulation results obtained in different environments.

4) Using pNMS, the methodology is validated by predicting the input–output functions experimentally characterized in skeletal muscles at various muscle lengths.

Finally, we discuss potential applications, limitations, and future directions for the current version of pNMS presented in this study.

## Methods

### Inverse modeling framework

Biophysical cell models were used as building blocks to model a neuromuscular system at cellular resolution (Fig 1). This building-block approach allows hierarchical construction and tractable analysis of population models for neuromuscular cells, including motoneurons (MNs), muscle–tendon fibers (MTs), and motor units (MUs).

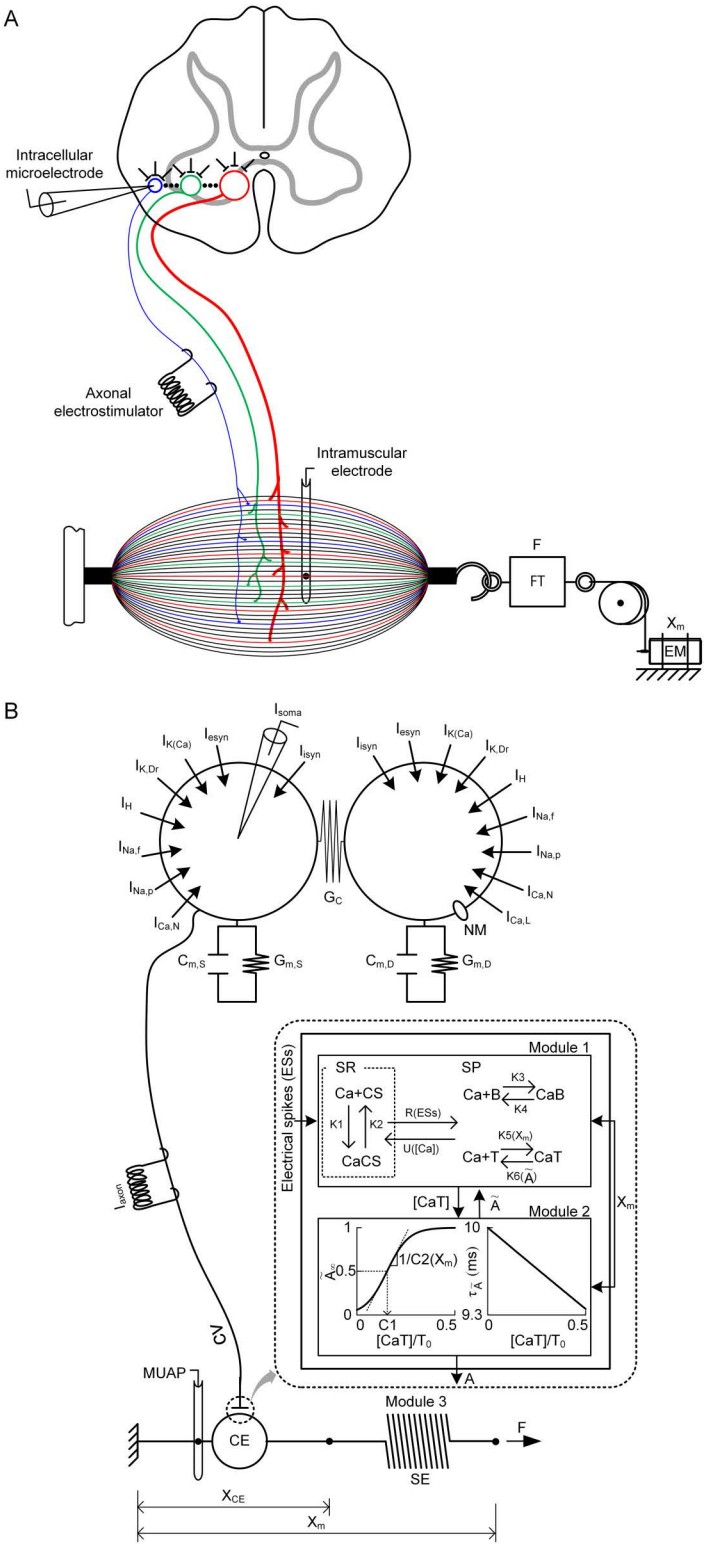

**Fig 1. Model description. A.** Schematic diagram of a typical experimental setting for primary neuromuscular cells. The upper panel shows a spinal cord cross-section with an intracellular microelectrode and an axonal electrostimulator. The lower panel shows a longitudinal section of the muscle–tendon

complex connected with a force transducer (FT) for force (F) measurement and an electric motor (EM) for length ($X_m$) control in series and an intramuscular electrode. The empty spheres and solid lines in the spinal cord indicate the motoneurons and their axons. The solid thin and thick lines in the muscle–tendon complex represent muscle fibers and tendons. The connections between a single motoneuron and muscle fibers (i.e., muscle unit) forming a motor unit are represented as filled circles in the muscle–tendon complex. The blue, green, and red colors indicate the S-, FR-, and FF-type motor units, respectively. **B.** Canonical model of a single motor unit. The upper and lower panels show the two-compartment model of the motoneuron and the three-modular model of the muscle–tendon unit. Two empty spheres represent the soma and dendrites in the motoneuron model. The solid black line between the motoneuron model and the muscle–tendon unit model indicates the axon of the motoneuron. The inset of the dotted circle area represents the transformation of the neural signals into the muscle activation level through two processes of calcium dynamics (Module 1) and cross-bridge formation (Module 2). The muscle force (F) and length ($X_{CE}$) were calculated via Hill mechanics, which consists of a contractile element (CE) and a serial elastic element (SE) (Module 3). The muscle unit action potentials (MUAPs) were produced using a typical MUAP shape that is experimentally measurable with an intramuscular electrode. The model parameters and system equations are fully presented in Appendix A in S3 Text.

We developed an inverse modeling framework to automatically determine the model parameters directly from the cell-specific properties (CPs) underlying essential features of cell behaviors that are experimentally observable. This study focused on the electrical properties of the MN membrane and the mechanical properties of the MT fiber. The model parameters were first classified into two groups. One group represents the range model parameters (RMPs) that effectively mediate the CPs and systematically vary across the population. In contrast, the other group represents constant model parameters (CMPs) that are consistent across the population. In this study, all RMPs and CMPs were biophysically relevant and categorized for modeling purposes. Then, we derived the inverse functions ($f^{-1}$) for the RMPs given the CPs pseudoanalytically. At the cellular level, this modeling framework can be formulated as the following equation:

$$\overline{RMP_X} = f^{-1}\left(\overline{CP_X}\right) \ \ where \ X \in \{MN, \ MT, MU\}$$

At the population level, cell organization is defined by the distribution of cell indicators (CIs) across the population of single cells (i). The CI represents a cell property measured as a cell classifier, associated with a cell type. In this study, somatic input resistance and maximal isometric force were selected as cell indicators for MNs and MTs, as experimentally suggested. The systematic variation in the RMP vector across the population was determined as a function of CI based on the correlations (g) between the CP vector and CI that are identifiable in the experiments. This population modeling process can be formulated as the following equation:

$$\overline{RMP_{X,i}} = f^{-1}\left(\overline{CP_{X,i}}\right) \ \ where \ \overline{CP_{X,i}} = g\left(CI_{X,i}\right), \ X \in \{MN, \ MT, MU\}, \ i \in \{1, \cdots, N\}$$

## Motoneuron modeling

MN functions are determined by both membrane mechanisms and their distributions across the dendritic structure. This feature was reflected in a recently developed conductance-based two-compartment modeling framework consisting of the somatic and dendritic compartments coupled by a single conductance [26] (see Appendix A for system equations and parameters in S3 Text). The RMPs for the MNs were selected as the effective parameters and uniquely determined for the CPs measured under different experimental conditions. Table 1 shows the relationships between the RMPs and CPs identified for MN modeling in this study. The RMPs were inversely determined at three stages. First, the five passive RMPs ($G_{m,S}$, $C_{m,S}$, $G_{m,D}$, $C_{m,D}$, and $G_C$) were analytically determined from the five passive CPs ($R_N$, $\tau_m$, $VA_{SD}^{DC}$, $VA_{DS}^{DC}$, and $VA_{SD}^{AC}$) under fully anesthetized states without active mechanisms. The dendritic input resistance ($R_{N,D}$) was represented by the product of $R_N$ and the asymmetry ratio ($VA_{SD}^{DC}/VA_{DS}^{DC}$) of dendritic signal propagation [27,28]. Second, the two active RMPs ($G_{Naf,S}$ and $f_S$) related to linear firing behavior were adjusted to match two action potential properties ($I_{rheo}$ and $t_{AHP1/2}$) at the soma without the involvement of active dendritic mechanisms [30,31]. Finally, the three active RMPs ($G_{CaI,D}$, $G_{K(Ca),D}$,

**Table 1. CPs and RMPs for motoneuron modeling.**

| CP | RMP | Description | Condition |
|---|---|---|---|
| Input resistance ($R_N$)[i]<br>Membrane time constant ($\tau_m$)[i]<br>Voltage attenuation factor ($VA_{SD}^{DC}$) in the soma-to-dendrite direction with direct current[‡]<br>Voltage attenuation factor ($VA_{DS}^{DC}$) in the dendrite-to-soma direction with direct current[‡]<br>Voltage attenuation factor ($VA_{SD}^{AC}$) in the soma-to-dendrite direction for alternating current[‡] | $G_{m,S}$ | Membrane conductance at the soma | Passive for both the soma and dendrite |
| | $G_{m,D}$ | Membrane conductance at the dendrite | |
| | $G_C$ | Coupling conductance between the soma and dendrite | |
| | $C_{m,S}$ | Membrane capacitance at the soma | |
| | $C_{m,D}$ | Membrane capacitance at the dendrite | |
| Rheobase current ($I_{rheo}$)[§] | $G_{Naf,S}$ | Peak conductance of fast Na$^+$ current at the soma | Active only for the soma |
| Afterhyperpolarization duration at half amplitude ($t_{AHP1/2}$)[§] | $f_S$ | Percentage of free to bound Ca$^{2+}$ ions at the soma | |
| Persistent inward current magnitude ($PIC_{mag}$)[φ] | $G_{Cal,D}$ | Peak conductance of L-type Ca$^{2+}$ current at the dendrite | Active only for the dendrite |
| Persistent inward current decay ($PIC_{decay}$)[φ] | $G_{K(Ca),D}$ | Peak conductance of calcium dependent K$^+$ current at the dendrite | |
| Persistent inward current amplification by neuromodulation ($PIC_{amp}$)[φ] | $S_{nm}$ | Scaling factor of $G_{Cal,D}$ | |

[i]Fleshman et al. [34], [‡]Kim et al. [26], [§]Hochman and McCrea [30], and [φ]Lee and Heckman [35]

and $S_{nm}$) underlying nonlinear firing behavior (i.e., bistable firing) were specified to reflect three plateau potential properties ($PIC_{mag}$, $PIC_{decay}$, and $PIC_{amp}$) during voltage clamping at the soma in the presence of monoamines [32,33].

## Muscle-tendon fiber modeling

MT functions are determined by complex interactions among the intracellular calcium dynamics, crossbridge formation between thin and thick filaments, and force production through the tendon anchoring the muscle to the bone. To reflect this feature, a recently developed modular modeling framework was selected because it can analytically constrain the model parameters with experimental data obtained under a full physiological range of stimulation frequency and muscle length [23] (see Appendix A for system equations and parameters in S3 Text). The RMPs for the MTs were selected as the effective parameters and uniquely determined for the CPs measured under different experimental conditions. Table 2 shows the relationships between the RMPs and CPs identified for MT modeling in this study. The RMPs were determined at individual modules. In module 1 for calcium dynamics, two RMPs ($\tau_1$ and $\tau_2$) were initially adjusted to match the twitch dynamics. Then, two RMPs ($\varphi_1$ and $\varphi_3$) were determined to reflect the length-dependent changes in the twitch amplitude. In module 2 for crossbridge formation dynamics, six RMPs (C1i, C1n1, C1n4, C2i, C2n1, C2n4, and $\alpha_i$) were first adjusted to match the dynamic calcium–force relationship. Then, two RMPs ($\beta$ and $\gamma$) were specified to reflect the movement-induced force degradation. In module 3 for Hill mechanics, one RMP ($P_{0.5}$) was first determined to reflect the peak force produced during isometric contraction at the intermediate MT length. Two RMPs ($g_1$ and $g_2$) were then determined to match the length–force properties under full excitation. Four RMPs ($a_0$, $b_0$, $c_0$, and $d_0$) were specified to reflect the velocity–force properties under full excitation. Finally, one passive RMP ($K_{SE}$) was determined to match the serial elastic stiffness. Additionally, two RMPs ($A_{MUAP}$ and $L_{MUAP}$) were determined to reflect the muscle unit action potential (MUAP) response.

## Motor unit modeling

The MU, a single MN and innervated MTs, shows one-to-one excitation and contraction, neural signal transduction delay, and spike-twitch speed matching [39]. This feature was reflected by coupling the MN to the MT model with an axonal

**Table 2. CPs and RMPs for muscle-tendon fiber modeling.**

| CP | RMP | Description | Module |
|---|---|---|---|
| Twitch dynamics[i] | $\tau_1$ | Time constant for the rising phase of Ca release from sarcoplasmic reticulum | Module 1 |
| | $\tau_2$ | Time constant for the falling phase of Ca release from sarcoplasmic reticulum | |
| Length dependent twitch amplitude[‡] | $\varphi_1$ | Rate of activation decrease for MT lengths less than the intermediate length | |
| | $\varphi_3$ | Rate of activation increase for MT lengths greater than the intermediate length | |
| Calcium–force relationship[§] | $c1i$ | Initial value of the calcium concentration for half of the maximal force | Module 2 |
| | $c1n1$ | Saturation limit of $c1i$ | |
| | $c1n4$ | Time constant for $c1i$ dynamics | |
| | $c2i$ | Initial value of the steepness of the calcium–force curve at $c1i$ | |
| | $c2n1$ | Saturation limit of $c2i$ | |
| | $c2n4$ | Time constant for $c2i$ dynamics | |
| | $\alpha_i$ | Activation degradation for fluctuating $Ca^{2+}$ concentration | |
| Movement induced force degradation[‡] | $\beta$ | Length coefficient for movement induced force degradation | |
| | $\gamma$ | Velocity coefficient for movement induced force degradation | |
| Maximal isometric force[‡] | $P_{0.5}$ | Peak isometric force at the intermediate MT length over its full physiological range | Module 3 |
| Length–force property under full excitation[ϕ] | $g_1$ | MT length for the peak of the length–tension curve | |
| | $g_2$ | Width of the length–tension curve | |
| Velocity–force property under full excitation[ϕ] | $a_0$ | Coefficient for Hill shortening equation | |
| | $b_0$ | Coefficient for Hill shortening equation | |
| | $c_0$ | Coefficient for Massima lengthening equation | |
| | $d_0$ | Coefficient for Massima lengthening equation | |
| Serial elastic stiffness[i] | $K_{SE}$ | Collective stiffness for intracellular and extracellular elastic properties | |
| Intracellular electromyography[ᵠ] | $A_{MUAP}$ | Amplitude of muscle unit action potential | |
| | $L_{MUAP}$ | Duration of muscle unit action potential | |

[i]Burke [2], [‡]Kim et al. [23], [§]Gordon et al. [36], [ϕ]Brown et al. [37], and [ᵠ] Merletti and Farina [38]

nerve model that perfectly transmits the action potentials to the muscle fibers with a time delay [40,41]. All the RMPs were determined using the same method applied to the MN and MT models. Additionally, one RMP (CV) for the conduction velocity of the MN axon nerve was determined to reflect the transduction delay from the motoneuron to the muscle unit [42]. Three RMPs ($f_S$ for MN and $\tau_1$ and $\tau_2$ for MT) were also considered for spike-twitch speed matching between the motoneuron afterhyperpolarization duration and the muscle twitch dynamics.

## Population modeling

The $R_N$ was selected as a CI for the MN population because of its systematic correlations with MN CPs under passive [5] and active [43] dendritic conditions. The path length (i.e., $D_{path}$) from the soma to the location of the persistent inward current channel over the dendrites varied as a function of $R_N$ on the basis of previous studies [28]. Notably, the combination of $R_N$, $I_{rheo}$, $t_m$, and $t_{AHP1/2}$ has been proposed as a reliable predictor (i.e., 97%) of conventional cell types (i.e., S, FR, and FF) [5]. The cell organization of the MN population model was specified by the relationship between the MN number and $R_N$. The RMPs of individual MN models were then automatically constrained inversely based on the correlations of $R_N$ to individual MN CPs (see Appendix B in S3 Text).

For the MT population, $P_{0.5}$ was chosen as a CI because of its systematic correlations with MT CPs in cat MTs. Notably, muscle fibers can be grouped into three conventional types (i.e., S, FR, and FF) according to the twitch speed and peak force [2]. The cell organization of the MT population model was specified by the relationship between the MT number and

$P_{0.5}$. The RMPs of individual MT models were then automatically determined to reflect the correlations of $P_{0.5}$ and individual MT CPs [44] (see Appendix B in S3 Text).

The MU population was constructed under the assumption that the motoneuron is coupled with the same type of muscle fibers (i.e., muscle unit) [45]. On the basis of this assumption, $P_{0.5}$ for the MT model was determined at the muscle unit level as a function of $R_N$. As a result, the cell organization of the MU population model was specified by the relationship between the MU number and $R_N$. A population model for motor units can be automatically constructed so that the RMPs for individual motor unit models systematically vary as a function of $R_N$, reflecting the correlations between $R_N$ and individual MU CPs.

### Efficient parallel simulation

To reduce the simulation time, we devised a parallel computing environment that operates under various computer systems, including a single computer with multiple cores, high-performance clusters, and a computer network via the internet. Particularly for high-performance clusters connected via a network, the parallel environment was designed and implemented to be controlled at the management node without manually installing all required software libraries and packages at individual computational nodes. This feature was achieved via the virtual environment technique. This technique enables all the required libraries and packages to be installed in a virtual folder and shared across the computational nodes through the network file system during simulations. The architecture of the parallel environment developed in this study is presented in Fig 2

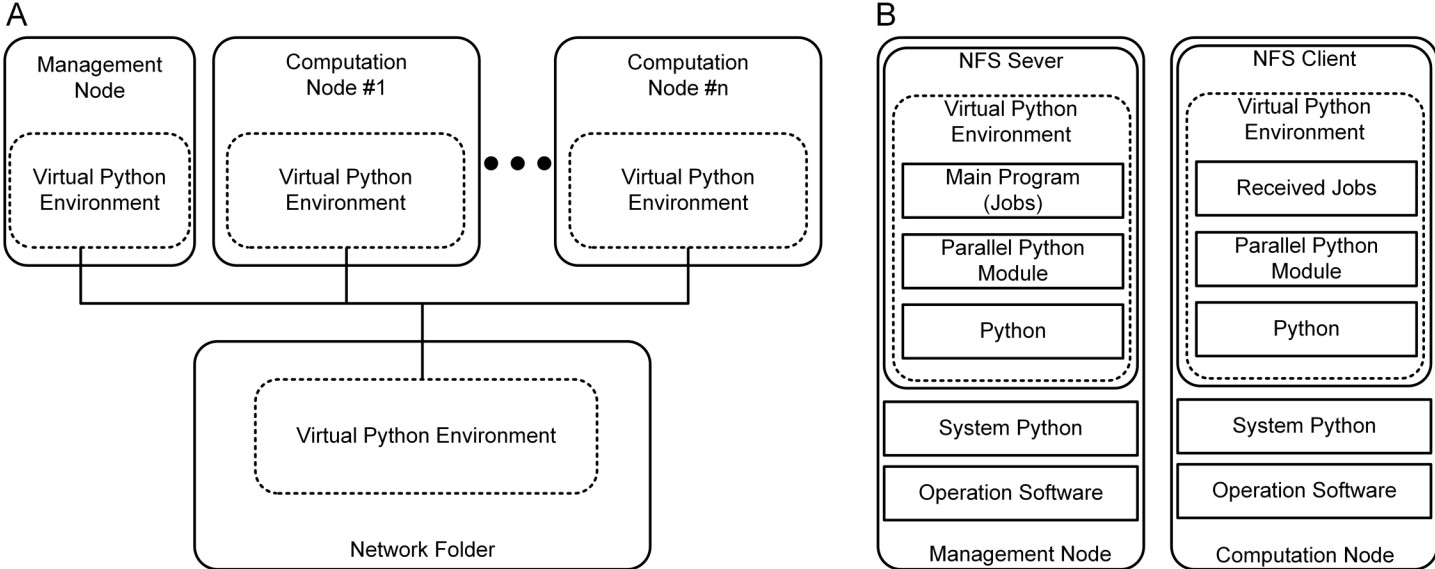

**Fig 2. Parallel environment. A.** Schematic diagram of the network architecture. Management and all computation nodes share the virtual Python environment that is installed within the shared network folder. The solid black lines indicate a typical network line connecting the management node, computation nodes, and network folder. **B.** Software components built in the management and computation nodes. The management and computation nodes function as network folder system (NFS) server and client sharing the same software components, such as the operation system, system Python, and virtual Python environment, where the parallel Python module and Python software and libraries are installed. The management node includes the main program, which distributes jobs to individual computation nodes via the network.

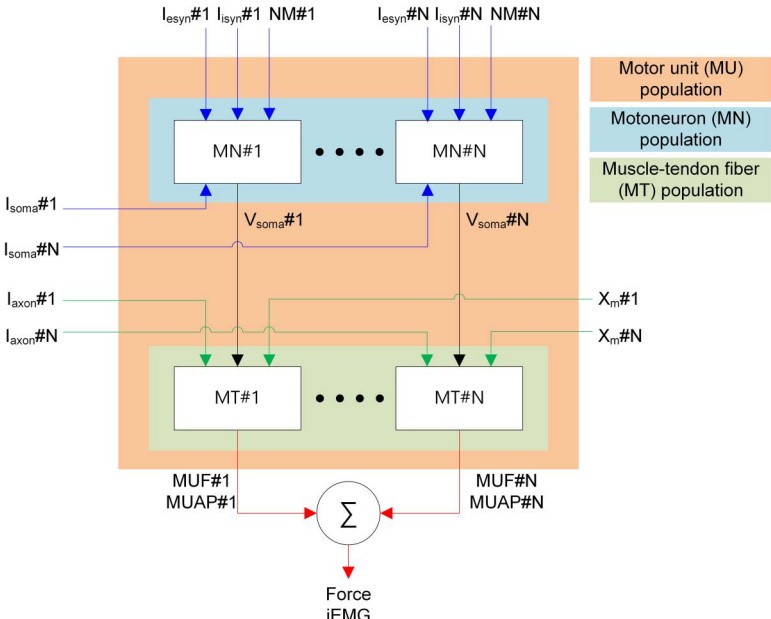

## Command-line software tool

To test our methodology, we developed an open-source Python package called pNMS along with an application programming interface (API) for easy use by others. The methodology is well suited for object-oriented programming, which is known to be appropriate for software maintenance and expansion. Fig 3 shows the overall architecture for the simulations under the pNMS software environment. This diagram shows the types of cells, populations, and signals that flow at the cellular and system levels. Input signals were assigned to individual cell models. Similarly, the output signals were generated from individual cell models and transferred to their connected cell models. Force and MUAP signals from individual MTs were linearly combined to calculate the total force [46] and the intramuscular electromyography signal (i.e., iEMG) measurable from within the muscle [47].

In pNMS, the user first defines the cell type (i.e., MN, MT, or MU) and number (i.e., 1~N), resulting in the automatic construction of a homogeneous cell population with the same parameter values. The cell organization is then specified as the distribution of CIs over the range of cell numbers. By default, the CPs are determined by their correlations with the CI experimentally identified from healthy adult cats under normal conditions (see Appendix B in S3 Text). The RMPs are inversely determined via inverse equations and lookup tables based on the CPs (see Appendix B in S3 Text). The CMPs are initialized with the parameter files, including the values identified in previous cat experiments (S1 Data). Optionally, the user can forwardly update individual correlations between the CP and CI to reflect different or abnormal conditions (see Appendix B in S3 Text). The population behaviors under diverse input conditions are efficiently simulated at the single-cell level in a parallel computing environment (i.e., a multicore computer and a computer cluster). Optionally, the simulation results can be displayed online in multiple modes and saved in separate files for further analysis. We present this overall process in a sequential diagram (Fig 4) and the command-line functions (Table 3) that can be easily incorporated into any

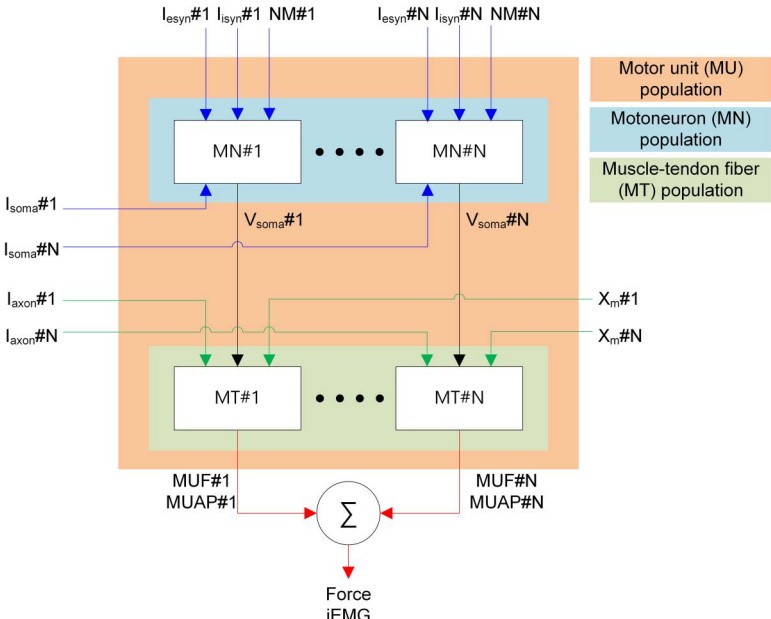

**Fig 3. Simulation architecture.** The white boxes indicate individual motoneurons and muscle–tendon fibers. The colored boxes indicate the pools of MNs, MTs, and MUs. Inputs (blue lines) to individual MNs include intracellular current injection at the soma ($I_{soma}$) and excitatory ($I_{esyn}$) and inhibitory ($I_{isyn}$) synaptic inputs and neuromodulatory inputs (NM) at the soma and dendrites. Inputs (green lines) to individual MTs include current impulse stimulation ($I_{axon}$) over the motoneuronal axon and MT length ($\mathbf{X_m}$). Outputs (black lines) from individual MNs include the action potential at the soma. Outputs (red lines) from individual MTs include the force (MUF) and action potential (MUAP) produced by each MU. The total force production and intramuscular electromyogram (iEMG) produced by the MU pool are estimated by summing the outputs of individual MTs.

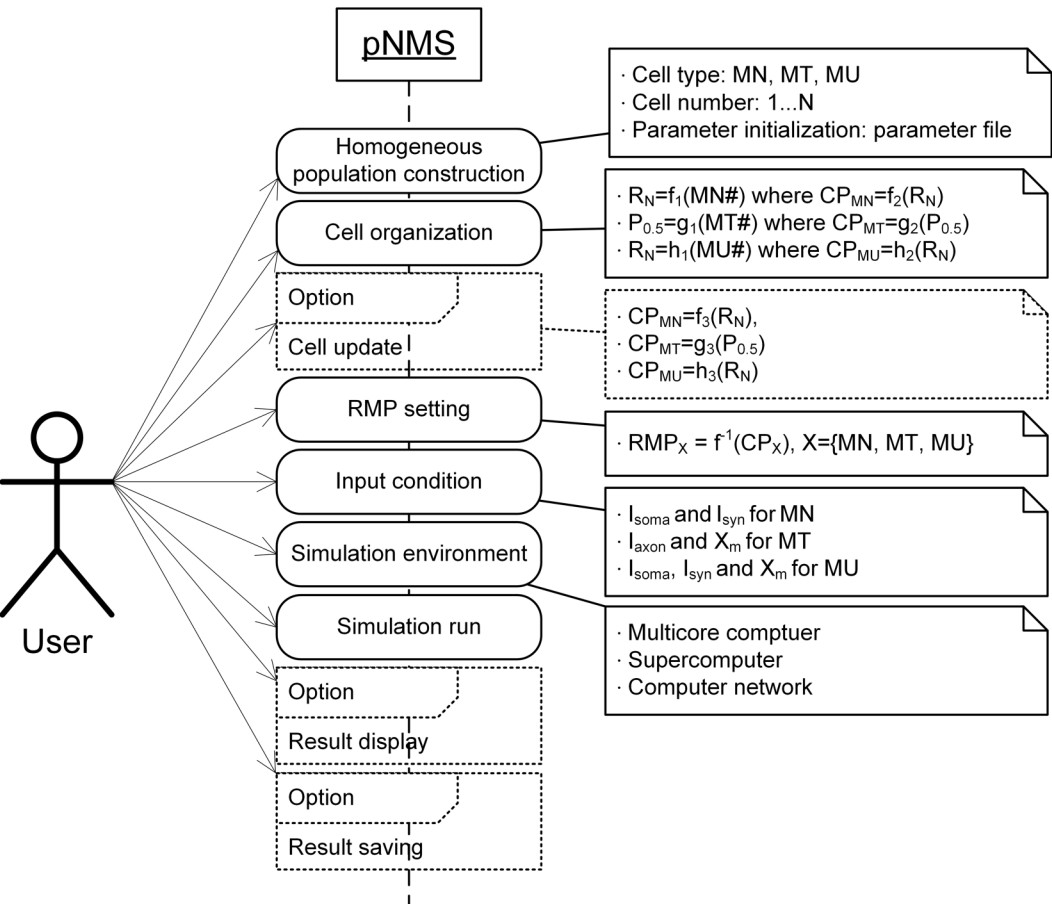

**Fig 4. User interface.** The solid and dotted boxes represent the necessary and optional operations the user can define in the pNMS environment. $R_N$ and $P_{0.5}$ indicate the input resistance measured at the soma of the motoneuron and the peak force measured from the muscle–tendon fiber during isometric contraction at the intermediate physiological muscle length, respectively. $RMP_X$ and $CP_X$ indicate the range model parameter and cell-specific property for cell type X, respectively. $I_{soma}$ and $I_{syn}$ indicate intracellular current injection at the soma and synaptic inputs in the motoneuron, respectively. $I_{axon}$ and $X_m$ indicate current impulse input over the axon and muscle-tendon length, respectively.

application program (see S2 Text for further details). Considerations for software development and instructions for parallel simulation are fully addressed in Appendices C and D in S3 Text.

## Results

### Cell population behavior

This section shows the feasibility of our methodology for reproducing the behavioral features of neuromuscular cells experimentally observed in healthy adult cats. The reliability and accuracy of the software were also verified by comparing the simulation results to previously published simulation results obtained via different software. The API functions used for this section are provided in parentheses in italics.

### Motoneuron population

We first constructed a homogeneous population model consisting of 10 fatigue-resistant (FR) motoneurons (MNs) with the same properties (*createPool*). Five active channels (i.e., $I_{Naf}$, $I_{Kdr}$, $I_{K(Ca)}$, $I_{Nap}$, and $I_{Can}$, along with dynamic variation in

**Table 3. Application programming interface (API).**

| Operation | API function | Description |
|---|---|---|
| Model Selection | createPool() | Creation of the homogenous population |
| Model Parameter Setting | setParameters() | Generation of range model parameter (RMP) values for the heterogeneous population |
| | plot_params() | Plotting the distribution of individual RMPs over the population |
| Simulation Condition Setting | setSimulTimes() | Setting the simulation time |
| | setInitialValues() | Setting the initial values for state variables |
| Input Parameter Setting | genNeuronInputSignals() | Generation of the intracellular current stimulation ($I_{soma}$) from the predefined functions |
| | importNeuronInputSignals() | Generation of the $I_{soma}$ from the data files |
| | setNeuronInputSignals() | Assignment of the $I_{soma}$ to individual motoneurons |
| | plotNeuronInputSignal() | Preview of the $I_{soma}$ |
| | genSynConSignals() | Generation of the synaptic conductance variation ($I_{syn}$) from predefined functions |
| | importSynConSignals() | Generation of the $I_{syn}$ from the data files |
| | setSynConSignals() | Assignment of the $I_{syn}$ to individual motoneurons |
| | plotSynConSignal() | Preview of the $I_{syn}$ |
| | genSpikeSignals() | Generation of the current impulse stimulation ($I_{axon}$) from the predefined functions |
| | importSpikeSignals() | Generation of the $I_{axon}$ from the data files |
| | setSpikeSignals() | Assignment of the $I_{axon}$ to individual muscle-tendon fibers |
| | plotSpikeSignal() | Preview of the $I_{axon}$ |
| | genMuscleLengthSignals() | Generation of the muscle–tendon length variation ($X_m$) from the predefined functions |
| | importMusscleLengthSignals() | Generation of the $X_m$ from the data files |
| | setMuscleLengthSignals() | Assignment of the $X_m$ to individual muscle-tendon fibers |
| | plotMuscleLengthSignal() | Preview of the $X_m$ |
| Parallel Condition Setting | setComputeNode() | Selection of the cores or computational nodes to be involved |
| Parallel Simulation | runSimulation() | Running the parallel simulation |
| Save and Display Condition Setting | saveSimulationResults() | Saving the simulation results to files |
| | plotSimulResult() | Online display of the simulation results |
| | plotImportData() | Offline display of the simulation results from the result files |

calcium reversal potential) were included in the soma to produce repetitive action potentials (MN_Parameters_3.0_FR.csv in S1 Data). In contrast, one active channel (i.e., $I_{Cal}$ and constant calcium reversal potential) was involved in the dendrite to generate a plateau potential. The values of the model parameters (*setParameters*), simulation time (*setSimulTimes*), and initial values for the state variables of the model equations (*setInitialValues*) were set to the same values used in a previous study [26]. Two types of input protocols were applied to show the bistable firing behavior of the motoneurons. A long-lasting step input with a brief (i.e., 50 ms) excitatory pulse followed by an inhibitory pulse with some delay (i.e., 10 sec) was applied to show the steady-state transition between two stable states, such as quiescent and regular firing or regular firing at different frequencies. Gradually rising and falling triangular input was used to show counterclockwise firing hysteresis, which is characterized by firing acceleration, firing saturation, and self-sustained firing behavior. These two input protocols were applied either to the somata (*genNeuronInputSignals* and *setNeuronInputSignals*) or to the dendrites (*genSynConSignals* and *setSynConSignals*). The peak of the triangular input corresponds to the synaptic excitation of 450% of full Ia afferent inputs over the dendrites in a realistically constructed FR-type motoneuron [48].

Parallel simulations were conducted on a multicore computer (*setComputNode* and *runSimulation*). The simulation results of the homogeneous MN population are shown in Figs A and B in S1 Text for current injection at the soma and

excitatory synaptic input over the dendrites (*plotSimulResult* and *saveSimulationResults*). All the models could replicate the bistable firing behaviors that were experimentally shown under step and ramp input conditions. These simulation results were consistent with those simulated under different software environments (i.e., XPPAUTO) [26]. Minor differences were observed and were likely caused by differences in the operating and hardware systems used.

Next, we systematically varied the $R_N$ and $D_{path}$ for the dendritic locations of the persistent inward current (PIC) channels across the MN population model using the built-in function (*setParameters*), as performed in a previous study [28]. The distribution of $D_{path}$ over the MU population was selected to match experimental observations from cats, such as voltage thresholds for PIC activation, their hysteresis, and input-output gains before full PIC activation. Fig 5 shows the RMPs (*plot_params*) automatically determined as a function of $R_N$ and $D_{path}$ based on the $R_N$–CP correlations characterized from adult cat MNs (see Appendix B in S3 Text). Notably, the RMP values do not necessarily reflect the values determined from the fully reconstructed MN models. The abrupt drop in $f_S$ (Fig 5D1) indicates the difference in half AHP duration ($t_{AHP1/2}$) between the slow- and fast-type motoneurons in adult cats [5]. The bell shape of $R_N$-$G_{Naf,S}$ curve (Fig 5D2) reflects the parabolic relation of $R_N$ and $I_{rheo}$ for cats [5], where motoneurons with low $R_N$ and low $I_{rheo}$ require more depolarizing inward current (i.e., $G_{Naf,S}$) to raise the membrane potential toward the threshold for an action potential. The remaining RMPs (i.e., $G_{K(Ca),D}$ and $S_{nm}$) were set to their default values (i.e., 0 and 1) across the population.

Fig 6 shows the systematic variation in the firing behavior of the MN models in response to slowly increasing and decreasing current stimulation at the soma (Fig 6A) and excitatory synaptic conductance over the dendrites (Fig 6B) across the population (*plotSimulResult* and *saveSimulationResults*). For both stimulation conditions, MNs with a lower threshold tended to show stronger nonlinearity in the firing response (i.e., firing acceleration and sustained firing below the threshold input) than those with a higher threshold. The population model could reproduce the simulated behaviors that were previously observed in different software environments (i.e., XPPAUTO) [28]. These results indicated that heterogeneity in cellular properties could considerably affect the output of the MN population.

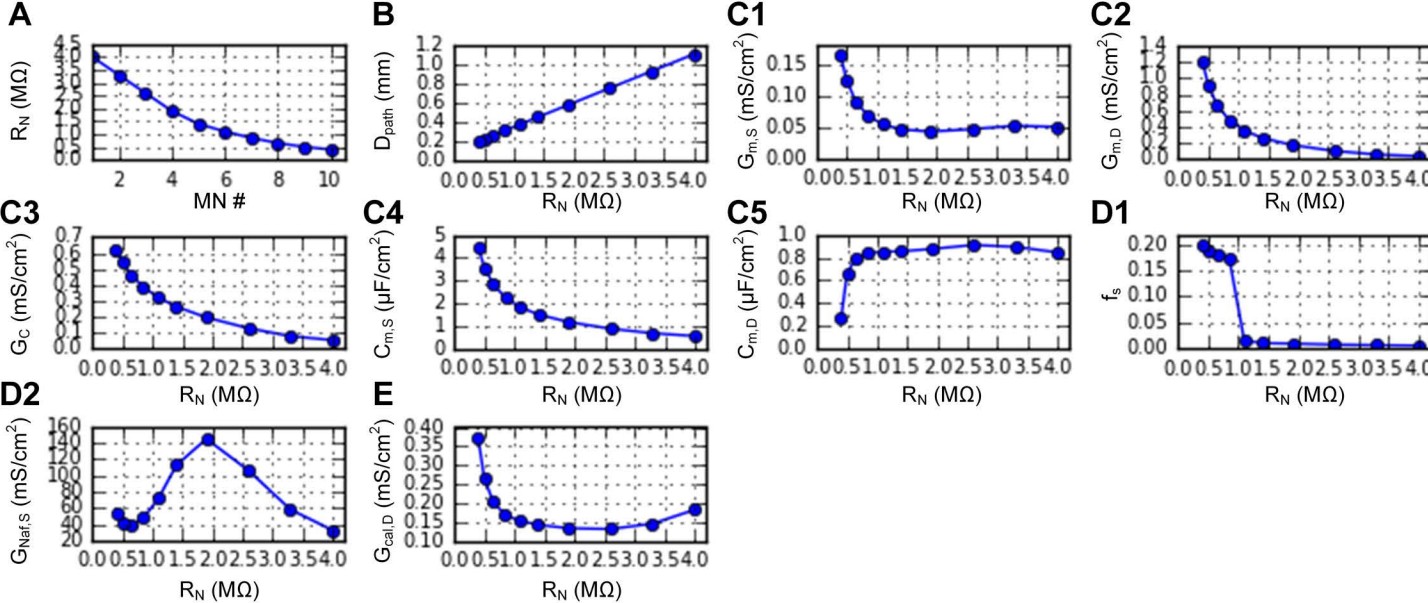

**Fig 5. RMP distribution for a heterogeneous motoneuron population. A.** Distribution of cell indicator. **B.** Distribution of dendritic PIC channels. **C1–C5.** Passive RMPs for electrotonic properties. **D1–D2.** Active RMPs for action potential properties. **E.** Active RMP for PIC amplitude.

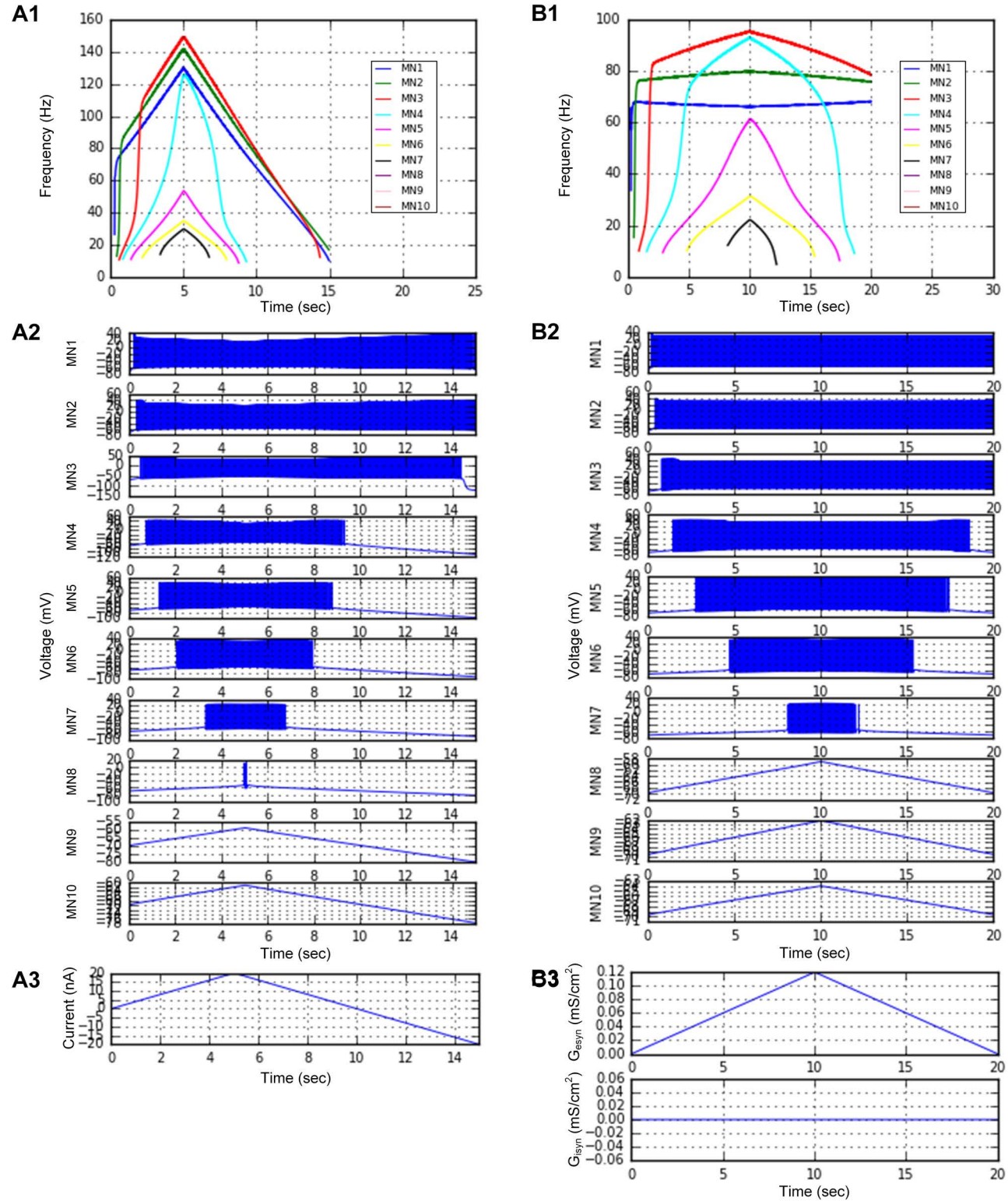

**Fig 6. Heterogeneous motoneuron population behavior.** A heterogeneous population model consisting of 10 MNs with different CPs, as specified in Fig 5, was constructed and simulated in a parallel environment. **A1–A3.** Instantaneous firing rates (**A1**) and voltage responses (**A2**) of the individual

motoneurons in response to the triangular current (**A3**) intracellularly injected at the somata. **B1–B3**. Instantaneous firing rates (**B1**) and voltage responses (**B2**) of individual motoneurons in response to excitatory synaptic inputs (**A3**) over the dendrites.

All instructions and codes for these simulations are presented in S2 Data for the homogeneous and heterogeneous MN populations.

## Muscle-tendon fiber population

We first constructed two homogeneous population models comprising 10 muscle-tendon fibers (MTs), one for slow-type MTs and the other for fast-type MTs (*createPool*). The values of the model parameters (MF_Parameters_3.0_SOL.csv and MF_Parameters_3.0_MG.csv in S1 Data, *setParameters*), simulation time (*setSimulTimes*), and initial values of the state variables (*setInitialValues*) were set to the same values for each population model as reported in previous studies [22,23]. Force was produced in response to the variations in the stimulation frequency (1~100 Hz) (*genSpikeSignals* and *setSpikeSignals*) and the MT length (isometric, isokinetic, and dynamic) (*genMuscleLengthSignals* and *setMuscleLengthSignals*), as applied experimentally and computationally. In this simulation, impulse current stimulation was applied to the muscle belly, and the conduction velocities were set to be the same for all the axonal nerves. The dynamic variation in MT length was generated to mimic the locomotor-like movement of adult cats (i.e., ~5 Hz).

Parallel simulations were conducted on a multicore computer (*setComputNode* and *runSimulation*). The simulation results for the homogeneous population model of slow and fast MTs are presented in Figs C and D in S1 Text (*plotSimulResult* and *saveSimulationResults*). Each MT population model could replicate the force production that was previously experimentally shown for the variations in the stimulation frequency and MT length [22,23]. Twitch, unfused, and fused contractions under isometric conditions showed a nonlinear relationship between stimulation frequency and muscle force (Figs Ca–Cd and Da–Dd in S1 Text for slow and fast MTs, respectively). Unfused contractions under isokinetic conditions demonstrated length- and velocity-tension properties (Figs Ce–Ch and De–Dh in S1 Text for slow and fast MTs, respectively). Unfused contractions under dynamic conditions indicated the dependence of force production on dynamic length variation (Figs Ci–Cj and Di–Dj in S1 Text for slow and fast MTs, respectively). These simulation results were consistent with those simulated under different software environments (i.e., NEURON) [22,23]. Minor differences were observed and were likely caused by differences in the operating and hardware systems used.

Next, we systematically varied $P_{0.5}$ using the built-in function (*setParameters*). Fig 7 shows the RMPs (*plot_params*) automatically determined from the linear correlations of $P_{0.5}$ with the CPs observed in adult cat muscle units (see Appendix B in S3 Text). Fig 8 shows the systematic variation in the force production of the heterogeneous MT models during twitch, unfused, and fused isometric contractions (Fig 8A–8D), fused isokinetic contractions (Fig 8E–8H), and unfused dynamic contractions (Fig 8I–8J) across the population (*plotSimulResult* and *saveSimulationResults*). Compared with those with lower $P_{0.5}$ values, the MT models with higher $P_{0.5}$ values presented greater force production and faster force development and cessation. The heterogeneous population model can predict the contributions of individual MT models to whole-muscle force production. Interestingly, in this population model, fast-type MTs showing sag behavior did not influence the overall shape of force production under partial excitation (i.e., 20~40 Hz) during isometric and dynamic contractions (see Fig 8C, 8D, and 8J). Notably, the rise in force before stimulation and the sustained force production after stimulation (i.e., MT10 in Fig 8A–8D) reflect the difference in calcium-activation relationship (i.e., Module 2 in Fig 1B) between the slow- and fast-type muscle fibers experimentally characterized in adult cats [22]. In addition, the distribution of axonal conduction velocity (Fig 7C) across the MT population model resulted in an orderly delayed onset and offset of force production. These results highlighted the impact of heterogeneity in the cellular properties on the output of the MT population.

All instructions and codes for these simulations are presented in S2 Data for the homogeneous and heterogeneous MT populations.

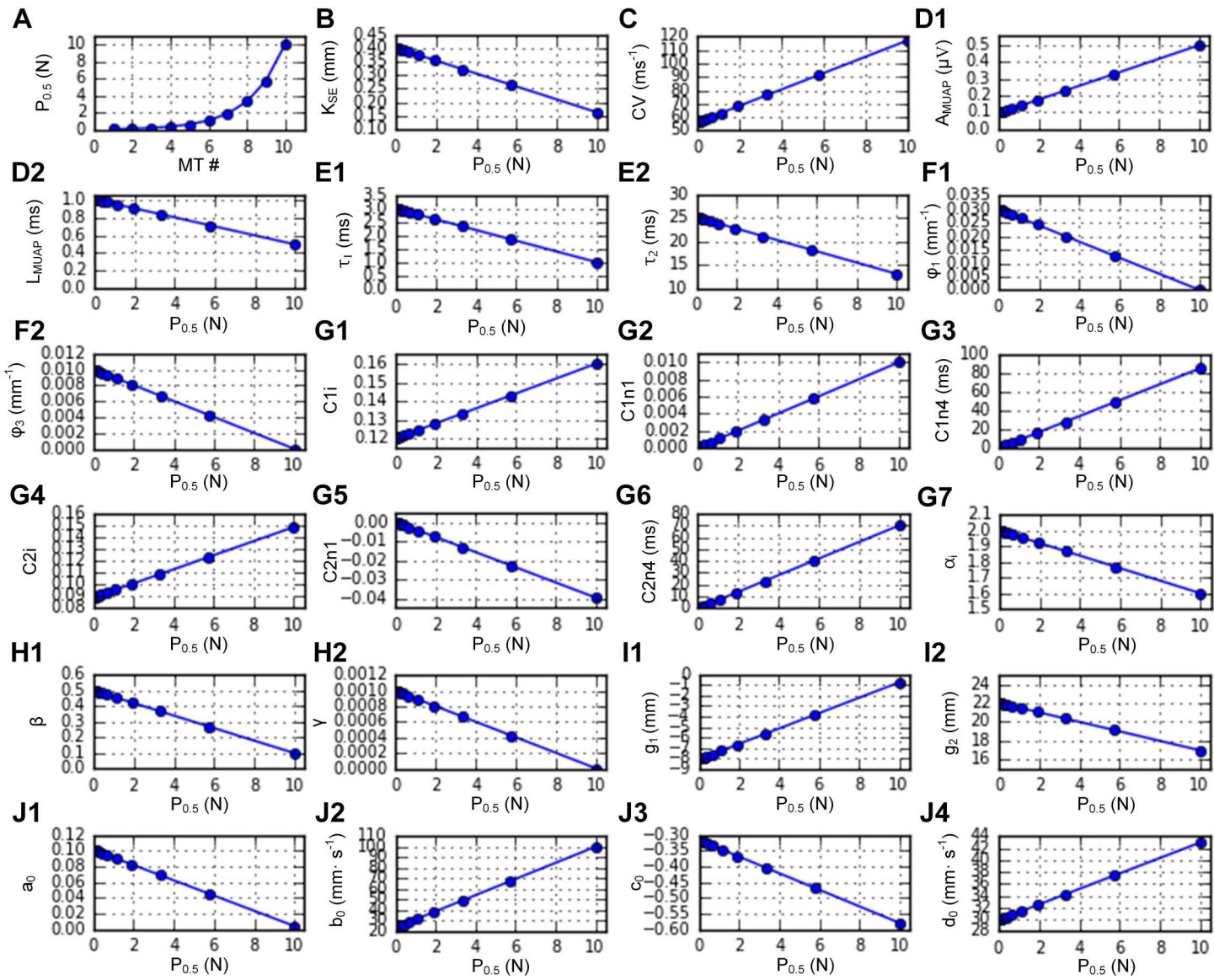

**Fig 7. RMP distribution for a heterogeneous muscle-tendon fiber population. A.** Distribution of cell indicator. **B.** RMP for the serial elastic stiffness. **C.** RMP for axonal nerve conduction. **D1–D2.** RMPs for the MUAP. **E1–E2.** RMPs for twitch dynamics. **F1–F2.** RMPs for the length-dependent twitch amplitude. **G1–G7.** RMPs for the calcium–force relationship. **H1–H2.** RMPs for movement-induced force degradation. **I1–I2.** RMPs for the length–force property under full excitation. **J1–J4.** RMPs for the velocity–force property under full excitation.

## Motor unit population

We first constructed a homogeneous population model comprising 10 slow-type motor units (MUs) (*createPool*). The same MN and slow MT models used in the previous sections were applied. The values of the model parameters (MN_Parameters_3.0_FR.csv and MF_Parameters_3.0_SOL.csv in S1 Data, *setParameters*), simulation time (*setSimulTimes*), and initial values of the state variables (*setInitialValues*) were set to the same values used in a previous study [49]. Two types of input protocols used in the previous section (Motoneuron population) were applied to show the input–output function of the slow-type MU population at the intermediate MT length ($X_{m,0.5}$). To evaluate the influence of the nonlinear firing

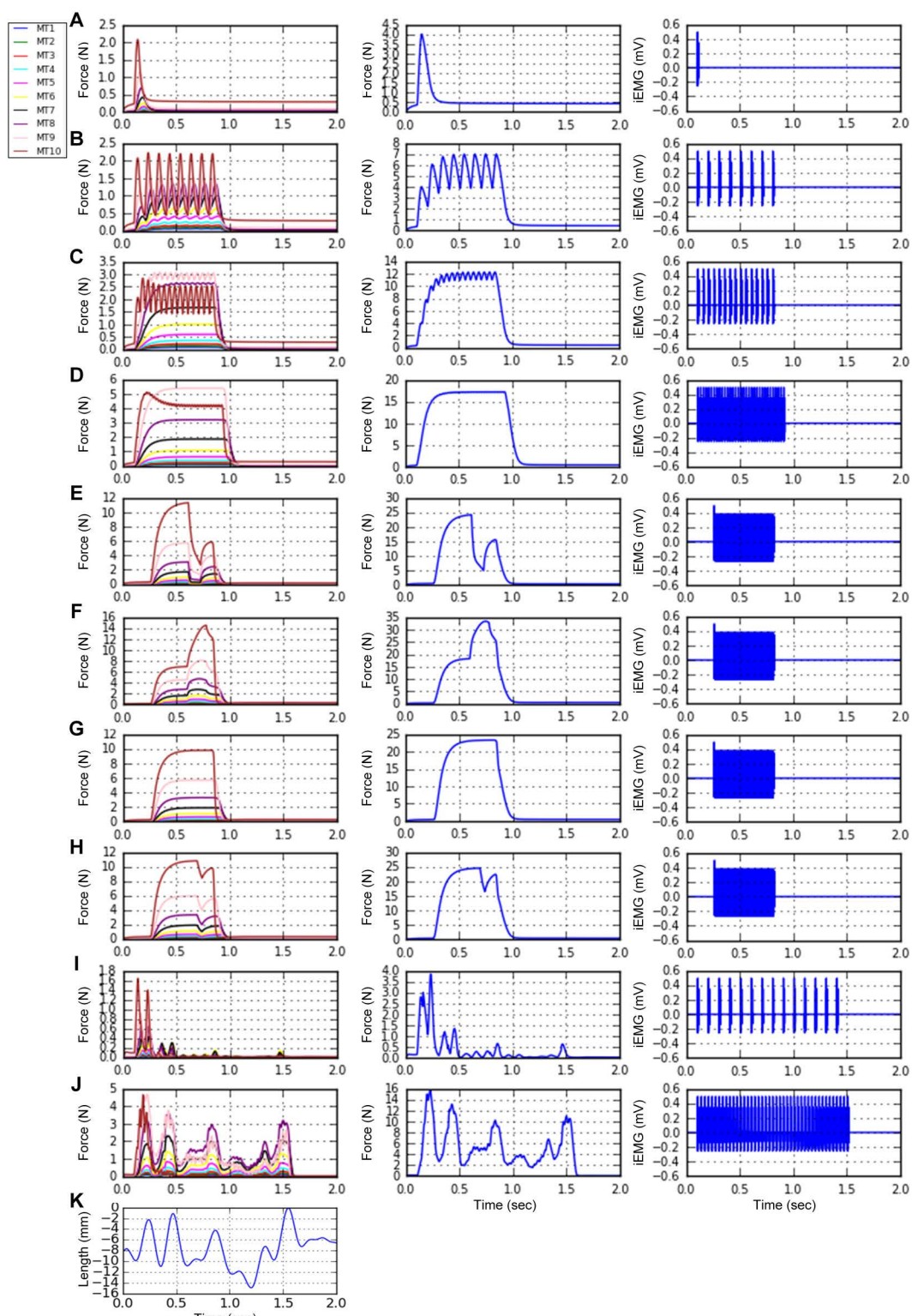

**Fig 8. Heterogeneous muscle-tendon fiber population behavior.** A heterogeneous population model consisting of 10 MTs with different CPs, as specified in Fig 7, was constructed and simulated in a parallel environment. **A–D**. Unfused tetani of individual MTs (left), total force production (middle),

and total electromyography measured at the center of the muscle unit territory (right) at constant stimulation frequencies (1, 10, 20, and 40 Hz for **A–D**, respectively) under isometric conditions at the intermediate length ($X_{m,0.5}$ = -8 mm). **E–H**. Fused tetani of individual MTs (left), total force production (middle), and total electromyography measured at the center of the muscle unit territory (right) under full excitation (100 Hz) during muscle shortening (**E**), lengthening (**F**), isometric contraction (**G**), and step shortening (**H**). **I–J**. Unfused tetani of individual MTs (left), total force production (middle), and total electromyography measured at the center of the muscle unit territory (right) during locomotor-like movement at constant stimulation frequencies (10 and 30 Hz for **I** and **J**, respectively). The time courses of current pulse stimulations and MT lengths applied for this figure are fully presented in Fig C in S1 Text. **K**. Muscle length variation applied to **I** and **J**.

behaviors (i.e., bistability and counterclockwise hysteresis with self-sustained firing) of MNs on force production by MTs during isometric contractions (*genMuscleLengthSignals* and *setMuscleLengthSignals*), step and triangular input protocols were applied to the MN soma (*genNeuronInputSignals* and *setNeuronInputSignals*) or dendrites (*genSynConSignals* and *setSynConSignals*), as reported in a previous study [49].

Parallel simulations were conducted on a multicore computer (*setComputNode* and *runSimulation*). The simulation results for the homogeneous population model of the slow MUs are shown in Figs E and F in S1 Text for current injection into the MN soma and excitatory synaptic input over the MN dendrites (*plotSimulResult* and *saveSimulationResults*). All the MU models demonstrated the nonlinear firing behaviors of the motoneurons and their consequences for the force production of the muscle units they innervate. These simulation results were consistent with those simulated under different software environments (i.e., PyMUS) [49]. Minor differences were observed and were likely caused by differences in the operating and hardware systems used.

Next, we systematically varied the $R_N$ across the MU population model, as performed in a previous study [29], where $D_{path}$ was set to a constant (i.e., 0.6 mm) (*setParameters*). An inverse relationship between $R_N$ and $P_{0.5}$ was further applied to couple the MNs and MTs. Additionally, a calcium-dependent potassium current ($I_{K(Ca),D}$) was added to the MN dendrites to reflect the PIC decay over time [43]. In this case, the default setting of $G_{K(Ca),D}$ to zero was redefined as a constant value (i.e., $G_{K(Ca),D}$ = 0.15 mS/cm$^2$) across the MU population to incorporate the CP of PIC decay. Resultantly, three RMPs (i.e., $f_S$, $G_{Naf,S}$, and $S_{nm}$) for the motoneuron remained unchanged when compared to the MN population in the previous section (Fig 5). Fig 9 shows the distributions of individual RMPs (*plot_params*) automatically set over the MU population.

Three schemes of synaptic conductance inputs were applied over the MN dendrites, considering three types of inhibitory inputs: background, recurrent, and reciprocal inhibition [29]. The resulting force outputs and intramuscular electromyography signals were estimated at the cellular and system levels. Fig 10 shows the systematic variation in the MN firing and MT force of the individual MUs during isometric muscle contractions at $X_{m,0.5}$ (i.e., -8 mm) (*genMuscleLengthSignals* and *setMuscleLengthSignals*) in response to the following input conditions: triangular current injection at the MN somata without a noisy background (Fig 10A6) (*genNeuronInputSignals* and *setNeuronInputSignals*); triangular excitatory synaptic input combined with constant inhibitory synaptic input over the MN dendrites under a noisy background (Fig 10B6) (*genSynConSignals* and *setSynConSignals*); balanced excitatory and inhibitory synaptic input over the MN dendrites under a noisy background (Fig 10C6) (*genSynConSignals* and *setSynConSignals*); and push–pull excitatory and inhibitory synaptic input over the MN dendrites under a noisy background (Fig 10D6) (*genSynConSignals* and *setSynConSignals*) across the population. The push–pull organization describes a decrease in inhibition with increasing excitation or vice versa via reciprocal inhibition [50]. Notably, the peak conductance of excitatory synaptic input was increased by almost three times to ensure all MUs were recruited during the ascending stimulation phase.

The simulation results (*plotSimulResult* and *saveSimulationResults*) obtained from the heterogeneous MU population model revealed the coexistence of onion skin and reverse firing patterns in the MN firing outputs (Fig 10A1– 10D1). In this study, the onion skin pattern indicates that the firing rate of low-threshold motoneurons is higher than that of high-threshold motoneurons. In contrast, the reverse pattern represents the opposite of the onion skin pattern, in which the firing rate of low-threshold motoneurons is lower than that of high-threshold motoneurons after firing acceleration on the ascending phase. This population model also demonstrated the modulation of the firing pattern and input–output gain

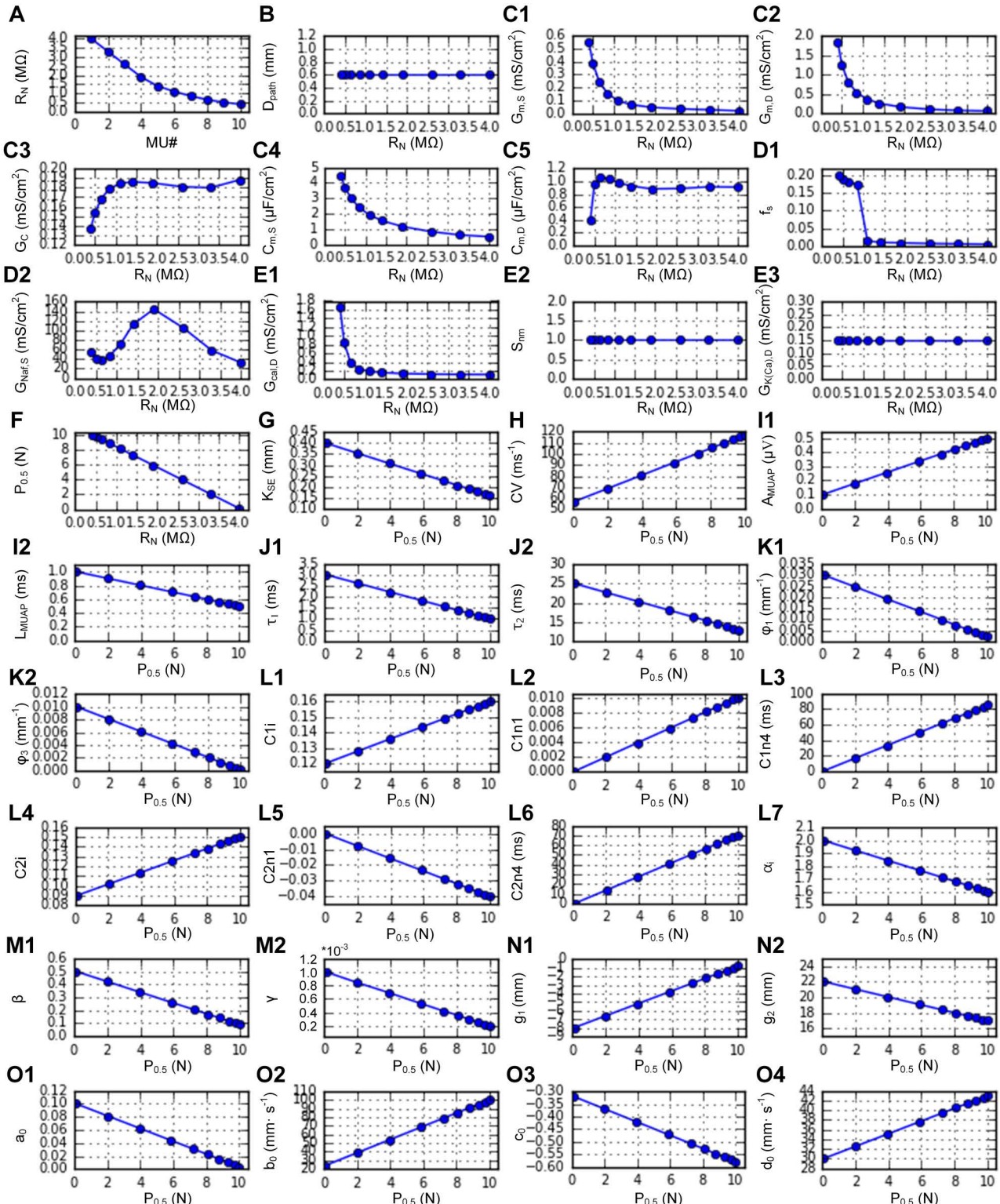

**Fig 9. RMP distribution for a heterogeneous motor unit population. A.** Cell organization of MNs. **B.** Distribution of dendritic PIC channels. **C1–C5.** Passive RMPs for electrotonic properties. **D1–D2.** Active RMPs for action potential properties. **E1–E3.** Active RMPs for plateau potential properties. **F.**

Cell organization of MTs. **G.** RMP for the serial elastic stiffness. **H.** RMP for axonal nerve conduction. **I1–I2.** RMPs for the MUAP. **J1–J2.** RMPs for twitch dynamics. **K1–K2.** RMPs for the length-dependent twitch amplitude. **L1–L7.** RMPs for the calcium–force relationship. **M1–M2.** RMPs for movement-induced force degradation. **N1–N2.** RMPs for the length–force property under full excitation. **O1–O4.** RMPs for the velocity–force property under full excitation.

of the MN population through the variation in the inhibitory synaptic input pattern. Accordingly, individual MTs can produce sequences of electromyographic signals (Fig 10A2–10D2) and forces (Fig 10A3–10D3). Interestingly, the FR-type MTs tended to produce the maximum force output. Furthermore, the shape of the positive outline of the total intramuscular electromyography signals (Fig 10A4–10D4) was closely related to that of the total muscle force (Fig 10A5–10D5). Overall, the MU population model could capture the nonlinear input–output characteristics that were evaluated using a realistic MU population model in a previous study [29].

All instructions and codes for these simulations are presented in S2 Data for the homogeneous and heterogeneous MU populations.

## Cell heterogeneity effects

This section predicted the effects of cell heterogeneity on population behavior by varying the cell organization at the MU level. A population model comprising 100 MUs was automatically constructed with the same parameter files applied for the previous MU population model. However, the linear $\tau_1$–$P_{0.5}$ relationship was updated to an exponentially decaying relationship derived from cat and human experiments (Fig 11A) [2,51]. The cell organization of the MU population model was modulated by changing the shape of the $R_N$–MU number relationship (Fig 11B). All the RMPs were inversely determined from the correlations between $R_N$ and CPs, as described in the previous section. The average firing pattern, total force production, and intramuscular electromyogram (iEMG) for individual MU population models were simulated under the same push–pull synaptic inputs for MNs and isometric conditions for MTs as in Fig 10D. We chose the push–pull scheme based on a computational study suggesting that it better fits the human data compared to the constant and balanced inhibition schemes [52]. This analysis was conducted in a parallel environment on a high-performance computing cluster with 80 cores (see Appendix D in S3 Text).

Fig 11 shows the simulation results for the dependence of population behaviors, such as the average firing rate, total muscle force, and root-mean-square iEMG envelope, on cell organization. The average firing rate decreased as the proportion of low-$R_N$ MUs (presumably F-type) increased (Fig 11C). In contrast, the magnitude of the total muscle force and iEMG envelope amplitude increased (Fig 11D and 11E). The relationship between the normalized iEMG and force tended to be linear in all MU population models (Fig 11F). Interestingly, the linearity of iEMG–force relation was more apparent for the linear $R_N$–MU number relationship representing a mix of low and high threshold motor units (yellow line in Fig 11F). The iEMG–force relationship became superlinear for the population models comprising more MUs with low $R_N$ (presumably F-type) and sublinear for those containing more MUs with high $R_N$ (presumably S-type). These results reflect that high threshold (low $R_N$) MUs need higher stimulation frequencies than low threshold (high $R_N$) MUs to produce the same level of muscle force because of the shorter contraction time of high threshold MUs. Notably, the iEMG–force relations were obtained after the delay in force development at the initial phase of synaptic inputs to the motoneurons (see the inset of Fig 11E). These results suggest that cell heterogeneity can play a crucial role in shaping the input–output functions of the motor unit population in the neuromuscular system.

We further investigated the dependence of the iEMG–force relationship on the muscle–tendon length. The population model involving the largest proportion of high-$R_N$ MUs (purple line in Fig 11B) was simulated under four isometric conditions at physiologically maximal (0 mm), intermediate (-8 mm), minimal (-16 mm), and further shortened (-24 mm) MT lengths. We injected a triangular current input (i.e., a peak of 40 nA at 5 sec over 10 sec) intracellularly at the somata

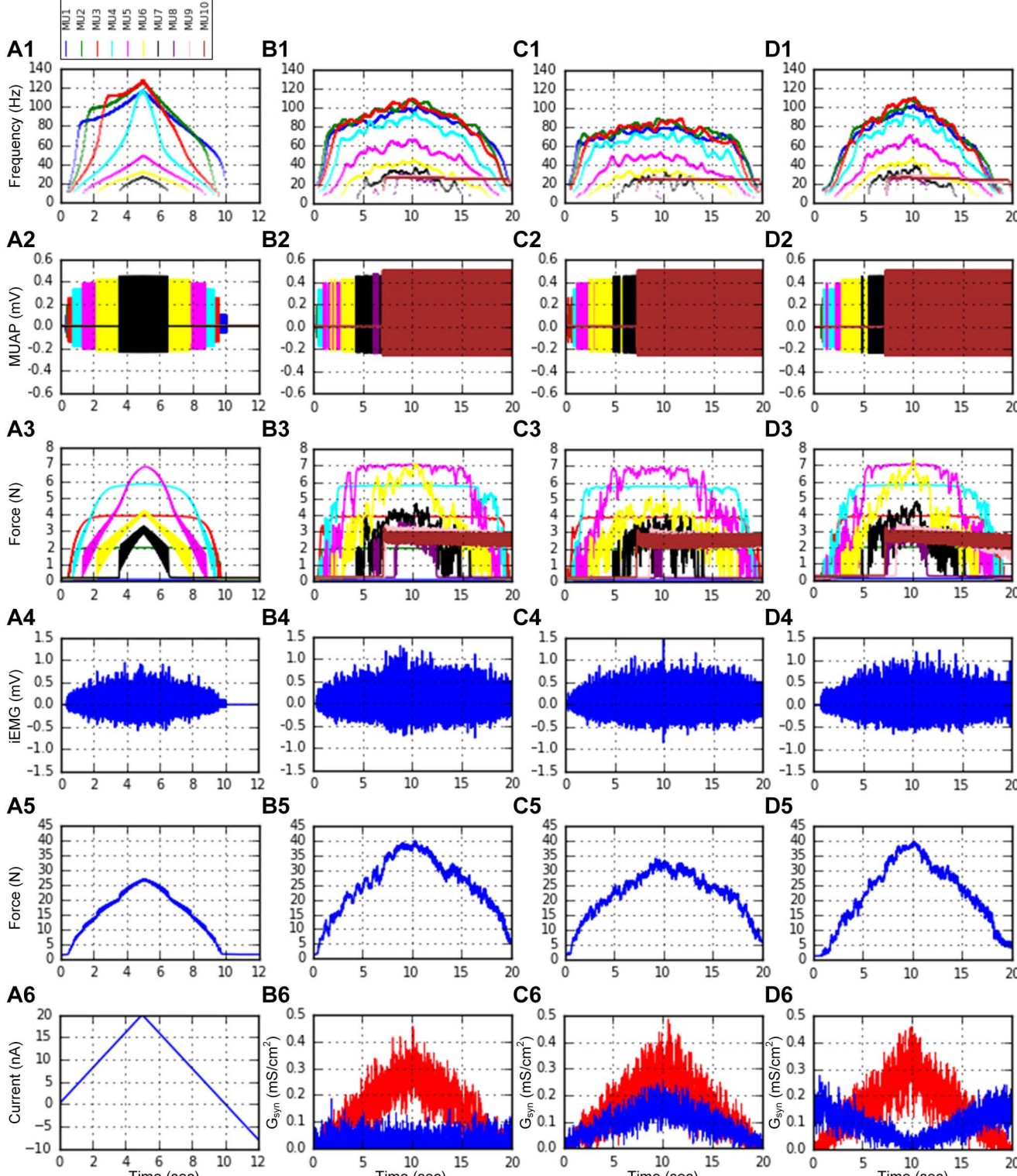

**Fig 10. Heterogeneous motor unit population behavior.** A heterogeneous pool model consisting of 10 MUs with different CPs, as specified in Fig 9, was constructed and simulated in a parallel environment. **A1–A6**. Instantaneous firing rates (**A1**), MUAPs (**A2**) and forces (**A3**) produced by the individual MUs, and the total intramuscular EMG (iEMG) (**A4**) and force (**A5**) produced by the MU population in response to triangular current injection (**A6**) to the somata of the MNs during isometric contractions at the intermediate MT length (-8 mm). **B1–B6**. Filtered instantaneous firing rates (**B1**), MUAPs (**B2**) and forces (**B3**) produced by the individual MUs, and the total iEMG (**B4**) and force (**B5**) produced by the MU population in response to excitatory

synaptic inputs (red) with background inhibitory inputs (blue) (**B6**) over the dendrites of the MNs during isometric contractions at the intermediate MT length (-8 mm). **C1–C6**. Filtered instantaneous firing rates (**C1**), MUAPs (**C2**) and forces (**C3**) produced by individual MUs, and the total iEMG (**C4**) and force (**C5**) produced by the MU population in response to a balanced pattern of excitatory (red) and inhibitory (blue) inputs (**C6**) over the dendrites of the MNs during isometric contractions at the intermediate MT length (-8 mm). **D1–D6**. Filtered instantaneous firing rates (**D1**), MUAPs (**D2**) and forces (**D3**) produced by the individual MUs, and the total iEMG (**D4**) and force (**D5**) produced by the MU population in response to the push–pull pattern of excitatory (red) and inhibitory (blue) inputs (**D6**) over the dendrites of the MNs during isometric contractions at the intermediate MT length (-8 mm). The raw data for **B1**, **C1**, and **D1** were smoothed with a Hanning filter and are presented in Fig G in S1 Text.

of motoneurons with passive dendrites for muscle contractions under full anesthesia without afferent feedback inputs. The peak of the current stimulation was chosen to ensure the recruitment of all motor units during the ascending phase of the triangular current injection. As a result, the time courses of the firing rate and EMG signal did not differ across muscle lengths (Fig 12A and 12B). However, the time course of force production depended on muscle length, reflecting the length-tension properties (Fig 12C). Consequently, the EMG–force relationship tended to be linear at the shortened length, but it became sublinear as the muscle length increased (Fig 12D). Notably, the iEMG–force relations were obtained by accounting for the delay (i.e., 0.5 sec) in force development at the initial phase of current injection to the motoneurons. These results support the hypothesis that the muscle length may alter the EMG–force relationship.

## Discussion

We proposed a methodology to automatically construct biophysically plausible and physiologically realistic population models of the primary cells comprising the neuromuscular system and to efficiently simulate and analyze them in parallel computing environments, including personal and networked computers. We implemented this methodology as a platform-independent Python package called pNMS, which includes user-friendly application programming interface functions to broaden its utility in the scientific community. Using pNMS, we demonstrated the contributions of cell heterogeneity to the input–output function of the motor unit population.

### Methodology for automatic modeling and efficient simulation

A forward modeling approach is typically applied where model parameters are numerically optimized to match the time courses of cell responses under different input conditions [53]. This approach, however, has shown a need for manual treatment for practical issues such as parameter redundancy, model instability, and simulation inaccuracy. A previous study further suggested that matching key cellular properties instead of cell outputs could reflect cell type-specific behaviors more effectively. Alternatively, we formulated an inverse modeling framework in this study. Under this framework, model parameters were uniquely determined directly from the cell-specific properties experimentally identified as key determinants underlying cell outputs under different input conditions. The cell organization was quantitatively modulated as a function of a cell indicator that is correlated with individual cell-specific properties. Combined with a building-block approach, this modeling framework enabled the automatic construction and tractable analysis of population models with various organizations of neuromuscular cells.

A parallel strategy is often employed to reduce the simulation time in large-scale simulations. However, a parallel computing environment is typically prepared by manually installing and updating software libraries necessary for simulations at individual computational nodes comprising a distributed computer system. This study resolved this issue by developing a parallel environment architecture based on a virtual environment and the network folder technique. All necessary libraries were installed and maintained in a virtual environment folder within a network folder shared among computational nodes. This architecture enabled the installation and deletion of required simulation libraries on all available computational nodes before and after simulation, controlled from the local management node.

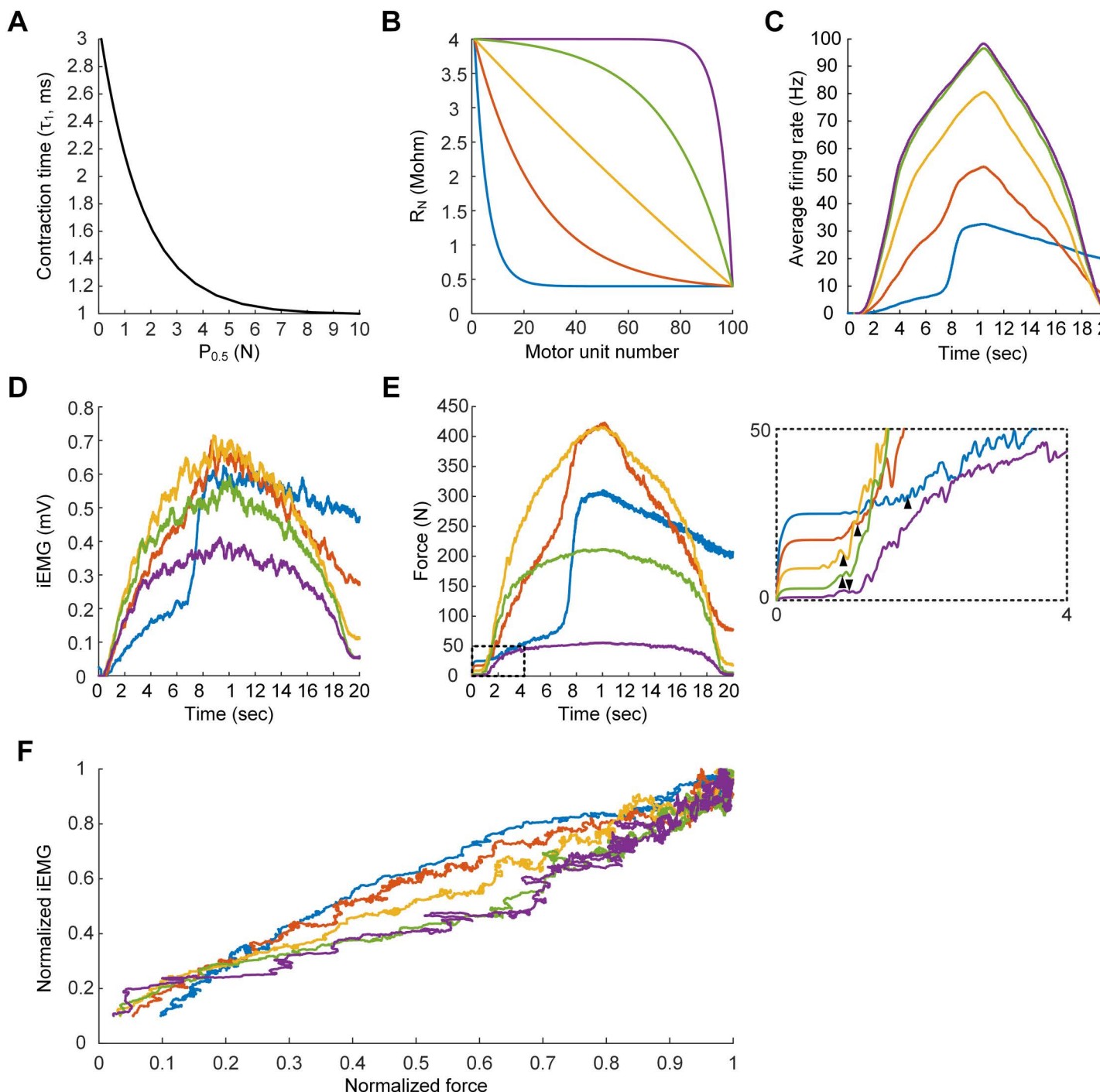

**Fig 11. Cell organization-dependent motor unit population behavior.** Five population models consisting of 100 MUs were constructed and simulated under the same input conditions used in Fig 10D. **A**. Contraction time–force production relationship applied to the MU population models. **B**. Cell organization patterns specified by the $R_N$–MU number relationships. **C**. Time courses of average firing rates of different MU populations. **D**. Time course of iEMG envelopes produced by different MU populations. **E**. Time course of total forces produced by different MU populations. The black arrowhead in the inset indicates the delay time in force production considered for the EMG–force relationship. **F**. iEMG–force relationships of different MU populations.

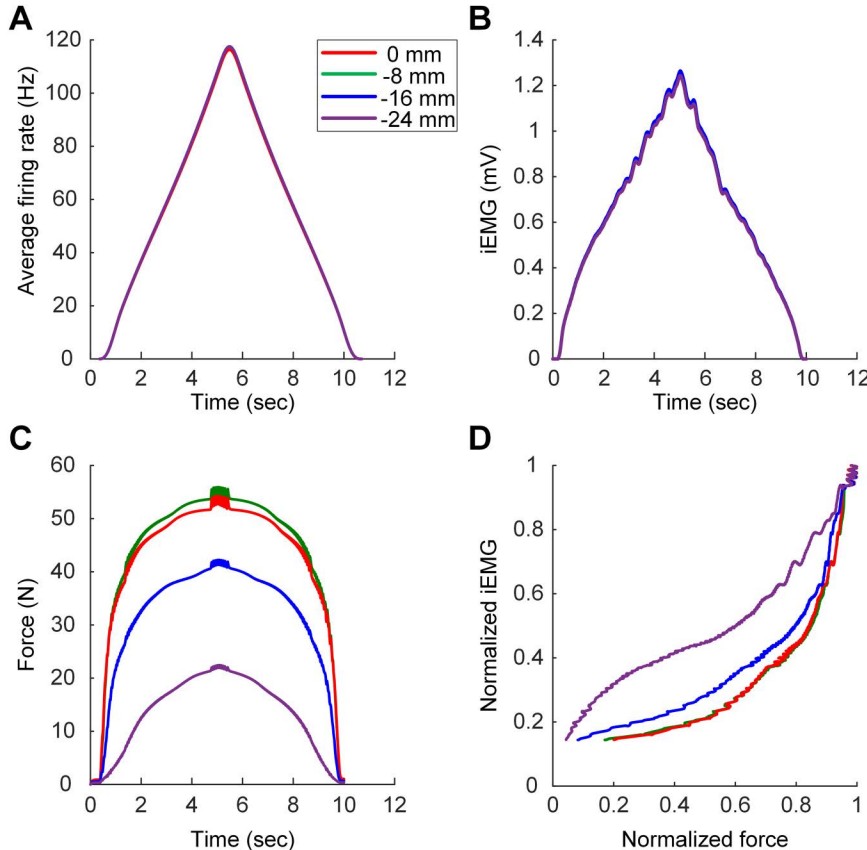

**Fig 12. Length-dependent EMG–force relationship.** The MU population model (purple line in Fig 11B) was simulated with passive MN dendrites in the absence of muscle afferent inputs under physiologically maximal (i.e., 0 mm), intermediate (i.e., -8 mm), and minimal (i.e., -16 mm) and further shortened (i.e., -24 mm) MT lengths. **A**. Time course of average firing rates at different MT lengths. **B**. Time course of iEMG envelopes at different MT lengths. **C**. Time course of total forces at different MT lengths. **D**. iEMG–force relationships at different MT lengths.

Our methodology supported object-oriented programming applied to large-scale software to improve maintenance, reusability, and extensibility [54]. Furthermore, the software tool (pNMS) implementing our methodology in this study provides command-line functions that can support step-by-step execution of the methodology and flexible incorporation into any application.

## Validation, reliability, and flexibility

Directly comparing pNMS to other software is difficult, mainly because of the discrepancy in the computational models and strategies that are employed. In this study, therefore, we verified the accuracy and stability of pNMS by reproducing the simulation results produced in different software environments used in previous studies. For example, the nonlinear responses of spinal motoneurons were simulated by pNMS and compared with those simulated by XPPAUTO software in a previous study [26,28]. The force production of the soleus and medial gastrocnemius muscles was simulated by pNMS and compared with that simulated by NEURON software in previous studies [22,23]. With respect to the motor unit, the input–output relationships of a single motor unit were simulated using pNMS and compared with those simulated via PyMUS software in a previous study [49]. The performance of pNMS was also assessed by measuring the real-time performance during the simulation of a neuromuscular system consisting of 10 motor units with different cellular properties

under noisy synaptic inputs over the motoneuron dendrites (Fig 10B). The real time needed for the simulation was much longer without parallel computing than with it on a multicore computer (e.g., 123 min vs. 38 min using a single core and eight logical cores, respectively, of an Intel Core i7-4790 CPU 3.6 GHz, for a 20-s simulation with 0.1 ms temporal resolution).

The implementation of user-defined custom models is currently limited in the pNMS environment. This limitation is partly due to the use of an inverse modeling approach that requires the analytical determination of model parameter values directly from target cell properties to automate population model construction. However, the current pNMS version provides flexibility in modifying the distribution of cell properties and relevant model parameters across the population model (Tables 1 and 2). For homogeneous population models, the user can change model parameter values in the Excel spreadsheet (i.e., model parameter files in S1 Data). For heterogeneous population models, the distribution of cell properties can be defined by users using built-in functions or importing user-defined data files (see Appendix B in S3 Text). Likewise, the user can also update the distribution of individual range model parameters using built-in functions or importing user-defined data files (see Appendix B in S3 Text). Furthermore, users can specify initial values for model equations, simulation time and resolution, parallel environment configuration, model inputs, and simulation outputs to plot online using built-in functions or user-defined data files. All these modeling and simulation settings can be performed via a command-line application programming interface (API) (Table 3 for API functions).

## Comparison to existing tools

Biophysical models for motoneurons have been simulated mostly using general-purpose software such as NEURON [25] and XPPAUTO [55]. In contrast, special-purpose software has been developed for muscle fibers [56] and motor units [44,57]. Typically, general-purpose software tools offer great flexibility in developing user-customized models, whereas special-purpose tools are limited to the canonical models embedded in their software. The current version of pNMS uses the canonical models for motoneurons and muscle fibers. However, unlike prior software tools, the canonical models are constrained directly by empirically measurable cell-specific properties. Furthermore, those models were designed and implemented as classes following the object-oriented programming principle, making maintenance and code extension easy and efficient.

To our knowledge, all existing software tools use a forward modeling scheme in which model parameters are treated as free parameters. Thus, users must optimize the model parameters to reflect the experimental data. However, the current version of pNMS offers both forward and inverse modeling modes, allowing the user to choose whether model parameters are analytically determined directly from experimental data or numerically optimized.

General-purpose tools have provided both graphical user interfaces and command-line interfaces (or application programming interfaces). In contrast, the special-purpose tools have provided only graphical user interfaces for setting the model and simulation conditions. The current pNMS version offers only a command-line interface, which is known to be efficient for simulation management and flexible for integration in broader applications. Future studies would be needed to design and develop a graphical user interface for pNMS, improving user accessibility and software usability.

Some software tools enable parallel simulation through a high-performance computing system or a web-based environment. The current pNMS version offers a flexible parallel simulation configuration, allowing users to run simulations on a single multicore processor, a high-performance computer cluster, or a web-based environment. Furthermore, the pNMS provides automated construction of a parallel environment without additional programming or library installation on computation nodes.

## Comparisons with previous studies

The fundamental elements of the neuromuscular system are the motoneurons and the muscle fibers they innervate. Multiple efforts have been made to construct computational models for populations of motoneurons, muscle fibers, and motor

units under various conditions. Under fully anesthetized conditions without neuromodulatory inputs from the brainstem, steady-state models for the motoneuron population were first developed mathematically to capture the recruitment current threshold, and the firing rate increased as the effective current level increased at the cell body [24,58]. The motoneuron dendrites were then explicitly modeled in a stick form, with one end attached to a ball representing the motoneuron soma exhibiting passive dendritic properties, to investigate the factors determining the response of human motoneurons to a composite excitatory input [59]. All these results could be replicated by setting the peak conductance of all ion currents to zero in the dendrites of the motoneuron model, as specified in the model parameter file within the pNMS environment.

To account for the neuromodulatory influence from the brainstem, persistent inward currents that mediate nonlinear firing behavior were incorporated into the dendrites of the motoneuron models while considering realistic morphology or a reduced single compartment. The reduced single-compartment modeling approach for active dendrites was applied in two-compartment motoneuron models [60,61]. pNMS could also capture the firing patterns of these motoneuron models by adjusting the parameters related to dendritic excitability under excitatory synaptic input over the dendrites (Figs A and B in S1 Text). Furthermore, one somatic compartment and four dendritic compartment models have been proposed to investigate the synaptic control of motoneuron firing patterns under a combination of excitatory and inhibitory synaptic inputs over the dendrites [29]. These results could also be captured by pNMS in response to the combination of excitatory and inhibitory synaptic inputs over the dendrites (Fig 10). Recently, the population of motoneuron models with fully dendritic structures revealed the coexistence of onion-skin and reverse onion-skin firing patterns [62]. pNMS can demonstrate this phenomenon in simulations of a heterogeneous population of motoneurons during triangular excitatory synaptic input over the dendrites (Fig 6).

To the best of our knowledge, there has been little empirical data on the relations of firing output and synaptic input across the motoneuron pool. Thus, this study used computational data from an anatomically reconstructed FR-type motoneuron model with a realistic distribution of synaptic inputs [48]. Previous research suggested that synaptic excitation of 450% of full Ia afferent inputs leads to the firing of FR-type motoneurons with active dendrites at up to 70 Hz. Our corresponding FR-type motoneuron model fired at a peak rate of 65 Hz with the same current injection at the soma, and at 48 Hz with the same synaptic input level to the dendrite (Figs A and B in S1 Text). Additionally, our motoneuron population matched the systematic variation in voltage threshold for PIC onset and two firing output-input intensity (F-I) gains before full PIC activation (i.e., F-I$_N$ for synaptic input and F-I$_S$ for current injection) across the motoneuron pool for cat hindlimb muscles, as shown in our previous study [28]. Furthermore, the low firing rate (< 40 Hz) observed during cat walking [63] is likely due to the combinatory effects of excitatory, inhibitory, and neuromodulatory inputs from various sources [64]. In this context, the pNMS may offer a flexible framework for systematically examining the impact of various intrinsic and extrinsic factors on the firing patterns of the motoneuron pool during locomotion.

For muscle modeling, a mathematical modeling approach was applied to simulate force production by the population of muscle fibers. Under isometric conditions, the muscle force was represented as a linear summation of twitch responses produced by individual muscle units at the cellular level [24]. To account for the dependence of force production on stimulation frequency and muscle length, a whole muscle model was developed to capture the experimental data obtained under various stimulation frequencies during isometric and isokinetic contractions [56]. All these results could be demonstrated using pNMS at both the cellular and whole-muscle levels under various conditions, including irregular stimulation frequency and dynamic muscle–tendon length variation (Fig 8).

Population models for the motor units were built by linking the mathematical model of the motoneuron pool developed under fully anesthetized conditions with the mathematical model of the muscle unit pool developed under isometric conditions [24,58]. To account for the neuromodulation effects from the brainstem, the two-compartment modeling approach was applied for motoneurons, along with the mathematical model of the muscle unit pool limited to the isometric condition in previous studies [60,61]. As a result of motor unit pool simulation, surface EMG signals were also produced in response to the firing patterns of motoneurons in previous studies [24,44]. Surface EMG signals measured from the skin electrodes

could be simulated through spatiotemporal filtering of the MUAP signals produced in the center of the muscle unit territory in our software environment [44]. All these results could be reproduced in the pNMS software environment by adjusting the model parameters properly (Fig 10).

Motor unit population models have been validated by matching the linear EMG–force relationship characterized during human experiments under isometric conditions [24]. Notably, previous motoneuron models have been limited to the fully anaesthetized state, in which their dendrites were assumed to be passive. In this study, the variability of the EMG–force relationship across different human muscles [65] was reproduced using motoneuron models with active dendrites under brainstem control via the pMNS software. The MU population models comprising a mix of low and high threshold moto- neurons tended to exhibit the linear relationship between EMG and force signal (yellow line in Fig 11). In contrast, the MU population models comprising mostly low (purple line in Fig 11) or high (blue line in Fig 11) threshold motoneurons tended to exhibit the nonlinear EMG–force relationship. Further work is needed to systematically investigate how the synaptic input profile and biophysical properties of motor units influence the EMG–force relations. Furthermore, the length depen- dence of the EMG–force relationship found in cat soleus muscles [66] could also be reproduced by MU population mod- els, which are composed mainly of low-threshold motoneurons in the pNMS environment (Fig 12).

Parallel simulation strategies have been proposed to realistically simulate the neuronal network system for the brain and spinal cord [44,67]. These parallel computations have been developed to target supercomputers or local computers connected through the internet. However, recent advances in multicore CPU technology have allowed parallel simulations on a single computer with multiple cores [68]. To facilitate the use of parallel computing, we developed the pNMS software to provide various options for parallel computing, including a single computer with multiple cores, a cluster system com- prising a network of computers, and distributed computers over the internet.

Previous models of the neuromuscular system have been used in various research fields, including neurophysiology, biomechanics, and biomedical engineering. However, physiologically plausible and computationally efficient software tool- kits publicly available for modeling and simulation of the neuromuscular system are highly desirable. The pNMS software developed in this study could further advance these fields by providing an open-source package that enables automatic construction and parallel simulation of biophysical models of the neuromuscular system across multiple levels and a wide range of experimental and physiological conditions [69].

## Limitations

All muscle fibers in a single MU were assumed to activate synchronously in the present study. However, the fibers within an MU can be activated asynchronously under abnormal conditions, such as fatigue, sarcopenia, or neuromuscular junction dysfunction. One way to simulate the asynchronous activation of muscle fibers in an MU in the current pNMS is to construct an MU population comprising identical MNs (i.e., same $R_N$) and MTs (i.e., same $P_{0.5}$) except for nerve conduction velocity (i.e., varying CV). Assigning different CV values by importing $P_{0.5}$–CV correlation data (Appendix B in S3 Text) can result in different time delays in the MUAP and force responses of individual MTs to the motoneuron spike in the pNMS. An example simulation and script were presented in Fig H in S1 Text and S2 Data.

This study also assumed that the same type of muscle fibers comprise an MU. However, individual muscle fibers within an MU may exhibit different mechanical properties. This scenario can be simulated by constructing an MU popu- lation comprising identical MNs (i.e., same $R_N$) but heterogeneous MTs (i.e., varying $P_{0.5}$). The $R_N$–$P_{0.5}$ correlation can be updated by importing user-specified data (Appendix B in S3 Text). An example simulation and script were presented in Fig I in S1 Text and S2 Data. Ultimately, the current MU model, which lumps all MTs into a single MT model, needs to be expanded to more realistically and precisely account for MT heterogeneity within an MU.

The inverse approach proposed in this study has been tested primarily for anesthetized, nonvoluntary cats. The reason was the lack of available data obtained during voluntary contractions. Thus, the current version of pNMS has an inher- ent limitation in predicting voluntary behaviors. However, the current pNMS software may serve as an efficient tool for

systematically investigating the input-output properties of the neuromuscular system at single-cell resolution. Moreover, future studies will validate the current pNMS software using empirical data obtained during voluntary control.

In addition, current experimental techniques are insufficient to measure the neuromuscular system's input-output behavior at single-cell resolution. Due to this limitation, this study evaluated whether the pNMS can capture the key properties of the neuromuscular system that have been experimentally characterized at the system level, such as EMG–force relations (Figs 11 and 12). Further experimental studies remain necessary to test the pNMS software against cellular-level input-output datasets for the neuromuscular system during voluntary contractions.

## Future directions

The current version of pNMS was developed in this study to model the neuromuscular system's foundational elements (motoneurons, muscle–tendon fibers, and motor units). Thus, pNMS must evolve to ensure the integrity of other biological elements contributing to the functioning of the neuromuscular system. For instance, the current version of the software package is limited to feed-forward modeling and simulation of the neuromuscular system. Thus, future studies need to extend the current feed-forward simulation to incorporate feedback signals. A variety of feedback signals have been observed in the neuromuscular system. Recurrent inhibitory feedback to motoneurons via Renshaw cells [70,71], excitatory interactions between motoneurons [72], and afferent sensory signals from the muscle-tendon complex [73] have been reported in the central nervous system and need to be included in future studies. With respect to motoneurons, variation in cell-specific properties, such as rheobase values at a given input resistance, should be considered for more physiologically accurate simulations. In muscles, fatigue [74,75], residual force enhancement [76], residual force depression [77], and activity-dependent potentiation [78] need to be considered when extending the muscle model. Furthermore, graphical user interfaces could be used to improve the user experience of the software. The current version of pNMS, which is based on an object-oriented programming paradigm and a unified modeling language, can also provide a basis for future extension by others, including chemical reactions observed in motoneurons and muscle fibers.

## Concluding remarks

This study presents a methodology and platform-independent open-source software tool (pNMS) to enable automatic population modeling and efficient parallel simulation of primary cells comprising the neuromuscular system under various situations. The traceability between the different levels of the neuromuscular system under the pNMS environment demonstrates the importance of cell heterogeneity in determining the population behaviors of the motoneuron, muscle–tendon fiber, and motor unit. The ability of pNMS to incorporate experimentally measurable cell properties may provide a practical computational solution applicable to various fields, from basic research on neuromuscular physiology to biomedical engineering for prosthetic technologies. pNMS is based on an object-oriented programming paradigm and a unified modeling language and can likely be used to develop an extensible platform suitable for community-based software development.

## Supporting information

**S1 Text. Supplementary figures and tables.**
(PDF)

**S2 Text. Application programming interface functions for pNMS.**
(PDF)

**S1 Data. Source codes of pNMS.**
(TAR)

**S2 Data. Python scripts for figures in the text and supporting information.**
(TAR)

**S3 Text. Appendices for model equations and parameters, inverse equations and lookup tables for the RMPs, considerations for software implementation, and instructions for parallel simulation.**
(PDF)

## Acknowledgments

The author truly thanks Jeonghwan Gwak, Yeongjae Kim, and Minjung Kim for their efforts in developing the early versions of the software. The author also thanks the reviewers for their valuable and constructive comments on the manuscript.

## Author contributions

**Conceptualization:** Hojeong Kim.

**Data curation:** Hojeong Kim.

**Formal analysis:** Hojeong Kim.

**Funding acquisition:** Hojeong Kim.

**Investigation:** Hojeong Kim.

**Methodology:** Hojeong Kim.

**Resources:** Hojeong Kim.

**Software:** Hojeong Kim.

**Validation:** Hojeong Kim.

**Visualization:** Hojeong Kim.

**Writing – original draft:** Hojeong Kim.

**Writing – review & editing:** Hojeong Kim.

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
