## [Decision Letter · Decision Letter 0]

16 Oct 2025

PCOMPBIOL-D-25-01612

Automating population construction and parallel simulation of biophysical models for neuromuscular cells

PLOS Computational Biology

Dear Dr. Kim,

Thank you for submitting your manuscript to PLOS Computational Biology. After careful consideration, we feel that it has merit but does not sufficiently meet PLOS Computational Biology's publication criteria as it currently stands. Therefore, we invite you to submit a revised version of the manuscript that addresses the points raised during the review process. Please pay particular attention to concerns raised by all three reviewers about the need for more independent experimental validation.

Please submit your revised manuscript within 60 days Dec 16 2025 11:59PM. If you will need more time than this to complete your revisions, please reply to this message or contact the journal office at ploscompbiol@plos.org. Please include the following items when submitting your revised manuscript:

* A response letter that responds to each point raised by the reviewers and describes what changes were made to the manuscript in response to each critique. You should upload this letter as a separate file labeled 'Response to Reviewers'. This file does not need to include responses to formatting updates and technical items listed in the 'Journal Requirements' section below.

We look forward to receiving your revised manuscript.

Kind regards,

Andrew D. McCulloch, Ph.D.

Academic Editor

PLOS Computational Biology

Pedro Mendes

Section Editor

PLOS Computational Biology

**Journal Requirements:**

1) Please upload all main figures as separate Figure files in .tif or .eps format. For more information about how to convert and format your figure files please see our guidelines:

2) We have noticed that you have uploaded Supporting Information files, but you have not included a list of legends. Please add a full list of legends for your Supporting Information files after the references list.

3) We notice that your supplementary information (Appendices) is included in the manuscript file. Please remove them and upload them with the file type 'Supporting Information'. Please ensure that each Supporting Information file has a legend listed in the manuscript after the references list.

Potential Copyright Issues:

i) Figures 1, and 4. Please confirm whether you drew the images / clip-art within the figure panels by hand. If you did not draw the images, please provide (a) a link to the source of the images or icons and their license / terms of use; or (b) written permission from the copyright holder to publish the images or icons under our CC BY 4.0 license. Alternatively, you may replace the images with open source alternatives. See these open source resources you may use to replace images / clip-art:

5) Please amend your 'Competing Interests' statement, and declare all competing interests beginning with the statement "I have read the journal's policy and the authors of this manuscript have the following competing interests:"

Note: If there are no competing interests to declare, please state "The authors have declared that no competing interests exist".

**Reviewers' comments:**

Reviewer's Responses to Questions

Reviewer #1: General Overview

This manuscript presents pNMS, a Python framework for building and simulating populations of motoneurons (MNs), motor units (MUs), and muscle-tendon fibers (MTs) with heterogeneous properties. Overall, the work is solid and clearly represents a useful tool for the neuromuscular modeling community. I appreciate the combination of open-source, object-oriented design, efficient parallel computing, and detailed inclusion of cellular features. The manuscript is well-written, logically structured, and provides enough methodological detail to reproduce the simulations, which makes it very accessible for other groups.

Novelty and Strengths

pNMS stands out for its use of inverse modeling to automatically set population parameters based on measurable cellular properties. This makes it possible to build heterogeneous MN–MT–MU populations in a systematic way. The model also captures detailed cellular-to-population dynamics, and its parallelized simulation on multicore systems or clusters makes it practical for larger-scale studies. Finally, being open-source and extensible in Python means it can be adapted, integrated, or expanded by other researchers, which is a big plus.

Major Comments

MUAP generation and fiber synchrony

Generating MUAPs for each individual fiber is a great feature, it lets you see how fiber-level differences affect the overall MUAP. That said, the current setup seems to assume all fibers in a MU fire synchronously. This could limit its usefulness for conditions where fibers don’t activate perfectly together, like fatigue, sarcopenia, or NMJ dysfunction. It would be good for the authors to discuss this limitation and maybe hint at ways it could be handled in future versions.

Heterogeneous fiber composition within a MU

The manuscript focuses on homogeneous fiber types within a MU, but it seems the framework could, in principle, mix fiber types. It would be helpful if the authors commented on how feasible this actually is, and whether there are limits, like computational cost, or issues with the inverse modeling framework if you combine fiber types without well-characterized CP correlations.

Stimulus protocol and translation to voluntary contractions

The validation is based on stimuli applied to anesthetized cats, so non-voluntary activation. It would be useful for the authors to clarify that using inverse modeling to predict voluntary contractions isn’t straightforward. This doesn’t have to be a flaw, but highlighting the intended scope of the model would help readers.

Validation with experimental data and inverse modeling

The paper emphasizes that inverse modeling ensures physiologically realistic parameter values. But this also introduces a conceptual issue: if you’re tuning parameters to match experimental outputs, then comparing your outputs to those same experiments isn’t really independent validation. The authors should acknowledge this limitation. True validation would either need independent measurements not used for parameter fitting, or testing whether the model can predict results under new, untested conditions.

Reviewer #2: General Comments

This was an ambitious, detailed effort to model most of the features influencing force and EMG production in mammalian muscle. Various conductances in both somatic and dendritic compartments of motoneurons were included in the model. Likewise, a large number of biophysical factors that shape motor unit force were included. These enabled simulations not only of isometric muscle force but forces arising while muscle changes length. Furthermore, the author provides some methods to handle the practical challenges of dealing with so many parameters. Indeed, this latter aspect seemed to be the main emphasis of the work. As such, this seems more a technical than scientific effort. Validation of the model output to previously published experimental work was meager and not that convincing. Furthermore, there were numerous minor concerns with the paper (detailed below).

Major Concerns

1. Throughout the manuscript, the author emphasizes that the outcomes using the present system match that developed on other platforms (e.g. Abstract “We validate the methodology using pNMS by comparing the simulation results with those from different software environments”). While this is important technically, it is of little scientific consequence. It is recommended that this matter be greatly downplayed.

2. While the EMG-force relations depicted in Fig. 11 F are roughly linear, they exhibit certain unphysiological features. In particular, they show an initial steep rise in EMG at low levels of force. This is opposite to what is found in some human muscles or otherwise the relation is strictly linear. The statement that “we reproduced the EMG–force relationships observed in cat” is without citation. Furthermore, it is difficult to imagine how one would obtain such an EMG-isometric force relation in the cat. The references to the work of Solomonow and and colleagues is not really relevant in this regard because they used electrical stimulation. Such stimulation causes all active motor units to discharge in synchrony. Such synchrony minimizes cancellation of partially overlapping motor unit potentials (and reduction of EMG signal at high intensities) that occurs naturally (e.g. Keena et al. Amplitude cancellation reduces the size of motor unit potentials averaged from the surface EMG. J Appl Physiol 2006). Regardless, the Solomonow paper (EMG-force relations of a single skeletal muscle acting across a joint: Dependence on joint angle. J Electromyogr Kinesiol 1991) also showed a slow rise in EMG with increased force at low contraction intensities – opposite to what was shown in the present simulations. As a consequence, the degree to which the population simulations were validated by experimental data is not strong.

Minor Concerns

1. Please embed figures and table in the text of the manuscript. Scrolling back and forth to the end of an 86 page manuscript is tedious for reviewers.

2. Why the focus on spinal motoneurons? Likewise, why restrict the approach to cat hindlimb muscles (Abstract “predict the population behaviors of neuromuscular cells in cat hindlimb muscles”). The utility of a modeling framework, such as advanced by the author, would be to use it flexibly to interrogate a wide range of motor unit populations, including those innervated by brainstem motoneurons and muscles throughout the body of various species.

3. (ln 130) “an inverse modeling framework has recently been proposed to overcome these issues by inferring and constraining effective model parameters directly from essential cell properties measured under various experimental conditions”. I would take some exception to “recently” as this seems to have been the modeling approach taken in biology for some time.

4. Why is Dpath greatest for neurons with greatest input resistance (e.g. Fig 5B). High input resistance cells would likely be those that are physically smaller, and as such, would seem to be expected to have a shorter Dpath.

5. Likewise, what us the physiological basis for the greater dendritic capacitance in the high input resistance cells (Fig. 5C5)?

6. What accounts for the abrupt drop in Fs (free calcium in soma, and linked to AHP duration) for neurons with input resistances > 1 Mohm? (Fig 5D1)

7. What is the intuition behind the inverted U shape in the relation between input resistance and fast sodium conductance (Fig. 5D2)?

8. Fig. 5A description “Cell organization” is too general and uninformative. Indeed, “cell organization” could apply to the entire Fig. 5.

9. Related to Figs. 5E2 and E3, there is no need to make a plot when all cells were assigned the same values. Just simply state that in the text.

10. Why in Fig 1 is a multi-channel intramuscular electrode array depicted? The simulation have only to do with one channel. How would the present framework simulate the same intramuscular activity but from different electrode perspectives?

11. (ln 376) “A step input with alternating brief excitatory and inhibitory pulses was applied to show the steady-state transition between two stable states” This is not clear. How can alternating depolarizing/hyperpolarizing pulses be used to reveal steady state behavior? What specifically do you mean by “brief”?

12. (ln 429) “Each MT population model could replicate the force production that was previously experimentally shown for the variations in the stimulation frequency and MT length”. Please provide citation to support this statement.

13. What accounts for the non-physiological behavior of motor unit 10 in Figure 8, namely the sustained force after stimulation has ceased? Likewise what account for the non-physiological initial small rise in twitches and tetany before the steep rise in force as the outset of stimulation?

14. What accounts for the non-physiological distributed EMG response to single stimuli in Fig. 8? Experimentally, such distributed responses are not seen even when stimulating the nerve quite distant from muscle.

15. Related to Figs. 8I and J, those are not “locomotor-like” movements. Locomotor movements are rhythmical and relatively brief. It would be helpful if the figures included what the input changes in length were associated with the force responses.

16. Fig. 9 seems to replicate most of what was already shown in Figs. 5 and 7. Therefore, there is no need to reshow those panels. Just mention what is different for the simulations of Fig. 10 from the parameters shown in Figs. 5 and 7. Also, if a parameter is not varied over the population, there is no need for a plot.

17. (lns 507-513). Please provide some physiological justification for the synaptic input schemes used. Also, define what is meant by “push-pull”.

18. (lns 516-517). Define the terms “onion skin” and “reverse firing”.

19. (ln 523). “the total intracellular electromyography signals” This should be intramuscular, not intracellular.

20. (ln 537) “τ1–P 0.5relationship was updated to an exponentially decaying relationship derived from cat experiments”. Cite reference to justify this statement.

21. (ln 543-545) “MU population models were simulated under the same push‒pull synaptic inputs for MNs and isometric conditions for MTs as in Figure 10 D”. Why this form of input? What is the physiological justification, if any?

22. There was no comment as to how reasonable the peak firing rates in the simulations shown in Fig. 10C were in comparison to recordings of cat hindlimb motor units. For example, Hoffer et al. (Cat hindlimb motoneurons during locomotion. II. Normal activity patterns. J Neurophysiol, 1987) rarely found firing rates in excess of 40 spikes/s across different speeds of treadmill locomotion.

23. The responses in the different motor units shown in Fig. 10 are confusing and unclear. Part of the problem is the use of the same color traces for different motor units. For example, there seems to be no force produced by motor unit 1 in panel A3. In panel B3, there appears to be a blue line that stays at zero force throughout. If that is MU 2, then why does it not get recruited. If it is MU 8, why does it not get recruited even though MU 10 appears to be activated. What accounts for the sustained activity in MUs 9 & 10 (panels B1, C1, D1) but not in other units?

24. In Fig. 10, panels A7, B7, C7, and D7 are unnecessary.

25. (lns 569-572) “0.0064 mS/cm 2 at the intermediate MT length (i.e., −8 mm), and 0.0128 mS/cm 2 at the lengthened MT length (i.e., 0 mm) under isometric conditions, as reported in a previous study.” Please provide citation that supports these afference conductances.

26. Legend to Figure 11. Change throughout “individual motor unit populations” to “different motor unit populations”

27. (ln 1036) “motor unit in human hindlimb muscles” should be “human leg muscles”

28. (lns 1052 – 1057) There are a large number of constants included in these equations without any explanation as to how they were determined. Likewise for ln 1079.

29. (ln 1772) “The lower panel shows a cross-section of the muscle‒tendon complex” Should be longitudinal, not cross – section.

30. Each table in the Methods should have a number and a title

31. For the table shown just after ln 1096, please indicate what parameter is represented in the table

32. The table just after ln 1102 is not needed. Likewise for the table following line 1107.

33. Each of the columns in the table after ln 1121 needs a label.

Reviewer #3: This is an interesting manuscript that introduces pNMS, an open-source software tool designed to construct biophysical population of reduced models of motor units (MUs). The tool provides pre-packaged parameter sets to construct motor unit’s population of heterogeneous properties, integrates a built-in simulation engine to solve the models ODEs, and supports parallel simulations. The author validated the tool results against general-purpose simulators such as NEURON and XPPAUTO. The work’s key value lies in its pre-packaged MU models and the integration of motoneuron, muscle-tendon, and whole motor unit dynamics within the same environment parallelization and scalability add practical advantages. However, several aspects require some clarification, particularly regarding usability, comparisons with existing tools, and definitions such as forward vs. inverse modeling. Specific comments are listed below.

---

The distinction between forward and inverse modeling frameworks is insufficiently explained in both the introduction and discussion. Both appear to use experimental data for parameter optimization, but how they differ in philosophy and application should be explicitly clarified. It is my understanding that some properties like the ion channel kinetics can be optimized once and then used with different conductance for different models - would that be considered forward or inverse modeling? Would framework such as Caillet et al. (2023) qualify as inverse modeling?

The author should clarify whether pNMS provides flexibility for users to implement custom models or is mainly tailored to pre-defined configurations. If flexibility is provided, it may be helpful to discuss how (and perhaps why) the user would accomplish this. Likewise, it may be helpful to add a section dedicated to how the user would modify/customize the simulations settings. I understand that changing the parameters on the excel sheets is one way.

The manuscript would greatly benefit from more explicit comparisons with existing tools. This includes comparisons with general-purpose simulators (NEURON, XPPAUTO, and Population/network-focused tools: NetPyNe) and MU-focused models/tools (Cisi & Kohn (2008), Elias & Kohn (2013, 2014)). Perhaps this could be achieved by splitting discussion Section 2 (Reliability and Efficiency) into two sections 1) Validation and reliability, 2) Comparison to existing tools. Also please elaborating on what you mean by “the discrepancy in the computational models” in line 631.

On the first page of Methods, too many terms are introduced at once (CPs, RMPs, CMPs, cell classifier, MT peak force). The author may consider adding more context or examples for these terms first. Also, please clarify how parameters are categorized as CMPs vs. RMPs—is this a modeling convention, or is it based on inherent properties (e.g., are ion channel kinetics CMPs or RMPs)?

The author my consider rewording lines 143-144 to “methodology that facilitates motor believe models developments and simulation” Instead of using “minimizes manual”. I believe manual tunning would be needed for any customization or new model.

The term “network folder” is unclear—please clarify whether this is standard terminology in parallel computing.

The term “neuromuscular cells” is not the most accurate here. I think the author is focusing on motoneurons specifically, and not muscle cells. I suggest using motoneurons. Or perhaps “motor units’ models”.

Line 91: The phrase “Computer modeling has recently been employed” is inaccurate. Modeling has a long history; the sentence would read better without “recently.”

Please clarify what you mean by using “mathematical tools” lines 98-99.

It would improve readability to avoid abbreviations in the headers, tables and figures, this will make it more readable (MT, MN, MU, etc)

The author may consider using less abbreviation if possible. Some terms like CIs, are only used very few times, so it would be better not to abbreviate terms used less frequently.

**Have the authors made all data and (if applicable) computational code underlying the findings in their manuscript fully available?**

The PLOS Data policy requires authors to make all data and code underlying the findings described in their manuscript fully available without restriction, with rare exception (please refer to the Data Availability Statement in the manuscript PDF file). The data and code should be provided as part of the manuscript or its supporting information, or deposited to a public repository. For example, in addition to summary statistics, the data points behind means, medians and variance measures should be available. If there are restrictions on publicly sharing data or code —e.g. participant privacy or use of data from a third party—those must be specified.requires authors to make all data and code underlying the findings described in their manuscript fully available without restriction, with rare exception (please refer to the Data Availability Statement in the manuscript PDF file). The data and code should be provided as part of the manuscript or its supporting information, or deposited to a public repository. For example, in addition to summary statistics, the data points behind means, medians and variance measures should be available. If there are restrictions on publicly sharing data or code —e.g. participant privacy or use of data from a third party—those must be specified.

Reviewer #1: Yes

Reviewer #2: Yes

Reviewer #3: Yes

PLOS authors have the option to publish the peer review history of their article (what does this mean?). If published, this will include your full peer review and any attached files.). If published, this will include your full peer review and any attached files.

.

Reviewer #1: No

Reviewer #2: No

Reviewer #3: **Yes:** Christopher K. ThompsonChristopher K. Thompson

**Figure resubmission:**
---

## [Decision Letter · Decision Letter 1]

31 Mar 2026

Dear Dr. Kim,

We are pleased to inform you that your manuscript 'Automating population construction and parallel simulation of biophysical models for neuromuscular cells: An inverse approach' has been provisionally accepted for publication in PLOS Computational Biology.

Best regards,

Pedro Mendes

Section Editor

PLOS Computational Biology

Reviewer's Responses to Questions

**Comments to the Authors:**

Reviewer #1: Authors addressed all concerns from the previous review. I would like to recommend this work for publication

Reviewer #2: The author is commended for assiduously addressing the concerns outlined in the previous review.

Reviewer #3: I thank the author for their thoughtful revision.

I would suggest the author would again consider what can be construed as an output form the motoneuron. I believe the author is talking exclusively about action potentials. It is still my opinion that the author refers to them as such, and not use the broader term "output".

**Have the authors made all data and (if applicable) computational code underlying the findings in their manuscript fully available?**

The PLOS Data policy requires authors to make all data and code underlying the findings described in their manuscript fully available without restriction, with rare exception (please refer to the Data Availability Statement in the manuscript PDF file). The data and code should be provided as part of the manuscript or its supporting information, or deposited to a public repository. For example, in addition to summary statistics, the data points behind means, medians and variance measures should be available. If there are restrictions on publicly sharing data or code —e.g. participant privacy or use of data from a third party—those must be specified.requires authors to make all data and code underlying the findings described in their manuscript fully available without restriction, with rare exception (please refer to the Data Availability Statement in the manuscript PDF file). The data and code should be provided as part of the manuscript or its supporting information, or deposited to a public repository. For example, in addition to summary statistics, the data points behind means, medians and variance measures should be available. If there are restrictions on publicly sharing data or code —e.g. participant privacy or use of data from a third party—those must be specified.

Reviewer #1: Yes

Reviewer #2: Yes

Reviewer #3: Yes

PLOS authors have the option to publish the peer review history of their article (what does this mean?). If published, this will include your full peer review and any attached files.). If published, this will include your full peer review and any attached files.

.

Reviewer #1: **Yes:** Alvaro Costa GarciaAlvaro Costa Garcia

Reviewer #2: **Yes:** Andrew J FuglevandAndrew J Fuglevand

Reviewer #3: **Yes:** Christopher K. ThompsonChristopher K. Thompson

---

## [Editor Report · Acceptance letter]

PCOMPBIOL-D-25-01612R1

Automating population construction and parallel simulation of biophysical models for neuromuscular cells: An inverse approach

Dear Dr Kim,

I am pleased to inform you that your manuscript has been formally accepted for publication in PLOS Computational Biology. Your manuscript is now with our production department and you will be notified of the publication date in due course.

With kind regards,

Anita Estes
